# Transferable Adversarial Attack based on Integrated Gradients

**Yi Huang, Adams Wai-Kin Kong**
School of Computer Science and Engineering
Nanyang Technological University
50 Nanyang Avenue, Singapore 639798
{yi.huangy,adamskong}@ntu.edu.sg

## Abstract

The vulnerability of deep neural networks to adversarial examples has drawn tremendous attention from the community. Three approaches, optimizing standard objective functions, exploiting attention maps, and smoothing decision surfaces, are commonly used to craft adversarial examples. By tightly integrating the three approaches, we propose a new and simple algorithm named Transferable Attack based on Integrated Gradients (TAIG) in this paper, which can find highly transferable adversarial examples for black-box attacks. Unlike previous methods using multiple computational terms or combining with other methods, TAIG integrates the three approaches into one single term. Two versions of TAIG that compute their integrated gradients on a straight-line path and a random piecewise linear path are studied. Both versions offer strong transferability and can seamlessly work together with the previous methods. Experimental results demonstrate that TAIG outperforms the state-of-the-art methods.[1]

## 1 Introduction

Adversarial example, which can mislead deep networks is one of the major obstacles for applying deep learning on security-sensitive applications (Szegedy et al., 2014). Researchers found that some adversarial examples co-exist in models with different architectures and parameters (Papernot et al., 2016b; 2017). By exploiting this property, an adversary can derive adversarial examples through a surrogate model and attack other models (Liu et al., 2017). Seeking these co-existing adversarial examples would benefit many aspects, including evaluating network robustness, developing defense schemes, and understanding deep learning (Goodfellow et al., 2015).

Adversarial examples are commonly studied under two threat models, white-box and black-box attacks (Kurakin et al., 2018). In the white-box setting, adversaries have full knowledge of victim models, including model structures, weights of the parameters, and loss functions used to train the models. Accordingly, they can directly obtain the gradients of the victim models and seek adversarial examples (Goodfellow et al., 2015; Madry et al., 2018; Carlini & Wagner, 2017). White-box attacks are important for evaluating and developing robust models (Goodfellow et al., 2015). In the black-box setting, adversaries have no knowledge of victim models. Two types of approaches, query-based approach and transfer-based approach, are commonly studied for black-box attacks. The query-based approach estimates the gradients of victim models through the outputs of query images (Guo et al., 2019; Ilyas et al., 2018; 2019; Tu et al., 2019). Due to the huge number of queries, it can be easily defended. The transfer-based approach, using surrogate models to estimate the gradients, is a more practical way in black-box attacks. Some researchers have joined these two approaches together to reduce the number of queries (Cheng et al., 2019; Guo et al., 2019; Huang & Zhang, 2020). This paper focuses on the transfer-based approach because of its practicality and its use in the combined methods.

---

[1]The code will available at https://github.com/yihuang2016/TAIG.

There are three approaches, standard objective optimization, attention modification, and smoothing to craft adversarial examples. In general, combining methods based on different approaches together yields stronger black-box attacks. Along this line, we propose a new and simple algorithm named Transferable Attack based on Integrated Gradients (TAIG) to generate highly transferable adversarial examples in this paper. The fundamental difference from the previous methods is that TAIG uses one single term to carry out all the three approaches simultaneously. Two versions of TAIG, TAIG-S and TAIG-R are studied. TAIG-S uses the original integrated gradients (Sundararajan et al., 2017) computed on a straight-line path, while TAIG-R calculates the integrated gradients on a random piecewise linear path. TAIG can also be applied with other methods together to further increase its transferability.

The rest of the paper is organized as follows. Section 2 summarizes the related works. Section 3 describes TAIG and discusses it from the three perspectives. Section 4 reports the experimental results and comparisons. Section 5 gives some conclusive remarks.

## 2 RELATED WORKS

**Optimization approach** mainly uses the gradients of a surrogate model to optimize a standard objective function, such as maximizing a training loss or minimizing the score or logit output of a benign image. Examples are Projected Gradient Descent (PGD) (Madry et al., 2018), Fast Gradient Sign Method (FGSM) (Goodfellow et al., 2015), Basic Iterative Method (BIM) (Kurakin et al., 2016), Momentum Iterative FGSM (MIFGSM) (Dong et al., 2018) and Carlini-Wagner's (C&W) (Carlini & Wagner, 2017) attacks. They are commonly referred to as gradient-based attacks. These methods were originally designed for white-box attacks, but are commonly used as a back-end component in other methods for black-box attacks.

**Attention modification approach** assumes that different deep networks classify the same image based on similar features. Therefore, adversarial examples generated by modifying the features in benign images are expected to be more transferable. Examples are Jacobian based Saliency Map Attack (JSMA) (Papernot et al., 2016a), Attack on Attention (AoA) (Chen et al., 2020) and Attention-guided Transfer Attack (ATA) (Wu et al., 2020b), which use attention maps to identify potential common features for attacks. JSMA uses the Jacobian matrix to compute its attention map, but its objective function is unclear. AoA utilizes SGLRP (Iwana et al., 2019) to compute the attention map, while ATA uses the gradients of an objective function with respect to neuron outputs to derive an attention map. Both AoA and ATA seek adversarial example that maximizes the difference between its attention map and the attention map of the corresponding benign sample. In addition to the attention terms, AoA and ATA also include the typical attack losses, e.g., logit output in their objective functions, and use a hyperparameter to balance the two terms. Adversarial Perturbations (TAP) (Zhou et al., 2018), Activation attack (AA) (Inkawhich et al., 2019) and Intermediate Level Attack (ILA) (Huang et al., 2019), which all directly maximize the distance between feature maps of benign images and adversarial examples, also belong to this category. TAP and AA generate adversarial examples by employing multi-layer and single-layer feature maps respectively. ILA fine-tunes existing adversarial examples by increasing the perturbation on a specific hidden layer for higher black-box transferability.

**Smoothing approach** aims at avoiding over-fitting the decision surface of surrogate model. The methods based on the smoothing approach can be divided into two branches. One branch uses smoothed gradients derived from multiple points of the decision surface. Examples are Diverse Inputs Iterative method (DI) (Xie et al., 2019), Scale Invariance Attack (SI) (Lin et al., 2020), Translation-invariant Attack (TI) (Dong et al., 2019), Smoothed Gradient Attack (SG) (Wu & Zhu, 2020), Admix Attack (Admix) (Wang et al., 2021) and Variance Tuning (VT) (Wang & He, 2021). Most of these methods smooth the gradients by calculating the average gradients of augmented images. The other branch modifies the gradient calculations in order to estimate the gradients of a smoother surface. Examples are Skip Gradient Method (SGM) (Wu et al., 2020a) and Linear back-propagation Attack (LinBP) (Guo et al., 2020). SGM is specifically designed for models with skip connections, such as ResNet. It forces backpropagating using skip connections more than the routes with non-linear functions. LinBP takes a similar approach, which computes forward loss as normal and skips some non-linear activations in backpropagating. By diminishing non-linear paths in a surrogate model, gradients are computed from a smoother surface.

## 3 PRELIMINARIES AND TAIG

### 3.1 NOTATIONS AND INTEGRATED GRADIENTS

For the sake of clear presentation, a set of notations is given first. Let $F : \mathbb{R}^N \to \mathbb{R}^K$ be a classification network that maps input $\boldsymbol{x}$ to a vector whose k-th element represents the value of the k-th output node in the logit layer, and $f_k : \mathbb{R}^N \to \mathbb{R}$ be the network mapping $\boldsymbol{x}$ to the output value of the k-th class, i.e., $\boldsymbol{F}(\boldsymbol{x}) = [f_1(\boldsymbol{x}) \cdots f_k(\boldsymbol{x}) \cdots f_K(\boldsymbol{x})]^T$, where $T$ is a transpose operator. To simplify the notations, the subscript $k$ is omitted i.e., $f = f_k$, when $k$ represents arbitrary class in $K$ or the class label is clear. $\boldsymbol{x}$ and $\widetilde{\boldsymbol{x}}$ represent a benign image and an adversarial example respectively, and $x_i$ and $\widetilde{x}_i$ represent their i-th pixels. The class label of $\boldsymbol{x}$ is denoted as $y$. The bold symbols e.g., $\boldsymbol{x}$, are used to indicate images, matrices and vectors, and non-bold symbols e.g., $x_i$, are used to indicate scalars.

Integrated gradients (Sundararajan et al., 2017) is a method attributing the prediction of a deep network to its input features. The attributes computed by it indicate the importance of each pixel to the network output and can be regarded as attention and saliency values. Integrated gradients is developed based on two axioms — *Sensitivity and Implementation Invariance*, and satisfies another two axioms — *Linearity and Completeness*. To discuss the proposed TAIG, the completeness axiom is needed. Thus, we briefly introduce integrated gradients and the completeness axiom below. Integrated gradients is a line integral of the gradients from a reference image $\boldsymbol{r}$ to an input image $\boldsymbol{x}$. An integrated gradient of the i-th pixel of the input $\boldsymbol{x}$ is defined as

$$\text{IG}_i(f, \boldsymbol{x}, \boldsymbol{r}) = (x_i - r_i) \times \int_{\eta=0}^{1} \frac{\partial f(\boldsymbol{r} + \eta \times (\boldsymbol{x} - \boldsymbol{r}))}{\partial x_i} \mathrm{d}\eta, \tag{1}$$

where $r_i$ is the i-th pixel of $\boldsymbol{r}$. In this work, a black image is selected as the reference $\boldsymbol{r}$.

**The completeness axiom** states that the difference between $f(\boldsymbol{x})$ and $f(\boldsymbol{r})$ is equal to the sum of $\text{IG}_i(f, \boldsymbol{x}, \boldsymbol{r})$, i.e.,

$$f(\boldsymbol{x}) - f(\boldsymbol{r}) = \sum_{i=1}^{N} \text{IG}_i(f, \boldsymbol{x}, \boldsymbol{r}). \tag{2}$$

To simplify the notations, both $\text{IG}_i(\boldsymbol{x})$ and $\text{IG}_i(f, \boldsymbol{x})$ are used to represent $\text{IG}_i(f, \boldsymbol{x}, \boldsymbol{r})$, and $\mathbf{IG}(\boldsymbol{x})$ and $\mathbf{IG}(f, \boldsymbol{x})$ are used to represent $[\text{IG}_1(f, \boldsymbol{x}, \boldsymbol{r}) \cdots \text{IG}_N(f, \boldsymbol{x}, \boldsymbol{r})]^T$, when $f$ and $\boldsymbol{r}$ are clear. The details of the other axioms and the properties of integrated gradients can be found in Sundararajan et al. (2017).

### 3.2 THE TWO VERSIONS — TAIG-S AND TAIG-R

We propose two versions of TAIG for untargeted attack. The first one based on the original integrated gradients performs the integration on a straight-line path. This version is named Transferable Attack based on Integrated Gradients on Straight-line Path (TAIG-S) and its attack equation is defined as

$$\widetilde{\boldsymbol{x}} = \boldsymbol{x} - \alpha \times \text{sign}(\mathbf{IG}(f_y, \boldsymbol{x})), \tag{3}$$

where the integrated gradients are computed from the label of $\boldsymbol{x}$, i.e., $y$, and $\alpha > 0$ controls the step size.

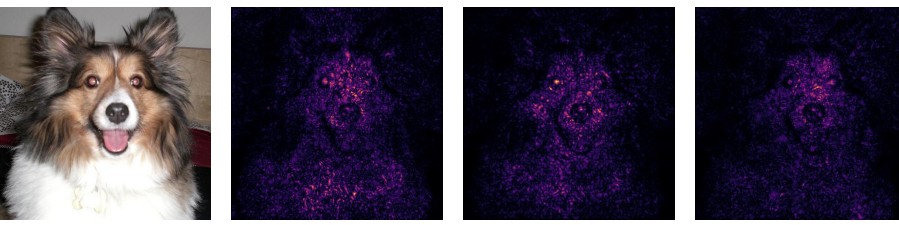

Figure 1: An original image and the corresponding integrated gradients from ResNet50, InceptionV3 and DenseNet121, from left to right.

The second version is named Transferable Attack based on Integrated Gradients on Random Piecewise Linear Path (TAIG-R). Let $P$ be a random piecewise linear path and $\boldsymbol{x}_0, \cdots, \boldsymbol{x}_E$ be its $E+1$ turning points, including the starting point $\boldsymbol{x}_0$ and the endpoint $\boldsymbol{x}_E$. The line segment from $\boldsymbol{x}_e$ to $\boldsymbol{x}_{e+1}$ is defined as $\boldsymbol{x}_e + \eta \times (\boldsymbol{x}_{e+1} - \boldsymbol{x}_e)$, where $0 \leq \eta \leq 1$. When computing integrated gradients of the line segment, $\boldsymbol{x}_e$ is used as a reference and the corresponding integrated gradients can be computed by equation 1. The integrated gradients of the entire path are defined,

$$\text{RIG}_i(f, \boldsymbol{x}_E, \boldsymbol{x}_0) = \sum_{e=0}^{E-1} \text{IG}_i(f, \boldsymbol{x}_{e+1}, \boldsymbol{x}_e). \tag{4}$$

The integrated gradients computed from the random piecewise linear path is called random path integrated gradients (RIG). Note that RIG still fulfills the completeness axiom:

$$\sum_{i=1}^{N} \sum_{e=0}^{E-1} \text{IG}_i(f, \boldsymbol{x}_{e+1}, \boldsymbol{x}_e) = \sum_{e=0}^{E-1} (f(\boldsymbol{x}_{e+1}) - f(\boldsymbol{x}_e)) = f(\boldsymbol{x}_E) - f(\boldsymbol{x}_0). \tag{5}$$

It should be highlighted that the integrated gradients computed from other paths also fulfill the completeness axiom (Sundararajan et al., 2017). In this paper, the turning points, $\boldsymbol{x}_e$ in the random path are generated by

$$\boldsymbol{x}_e = \boldsymbol{x}_0 + \frac{e}{E}(\boldsymbol{x}_E - \boldsymbol{x}_0) + \boldsymbol{v}, \tag{6}$$

where $e \in (0, 1, \cdots, E)$ and $\boldsymbol{v}$ is a random vector following a uniform distribution with support from $(-\tau, \tau)$. The attack equation of TAIG-R is

$$\widetilde{\boldsymbol{x}} = \boldsymbol{x} - \alpha \times \text{sign}(\mathbf{RIG}(f_y, \boldsymbol{x})), \tag{7}$$

which is the same as TAIG-S, except that $\mathbf{IG}(f_y, \boldsymbol{x})$ in TAIG-S is replaced by $\mathbf{RIG}(f_y, \boldsymbol{x})$. As with PGD and BIM, TAIG can be applied iteratively. The sign function is used in TAIG because the distance between $\widetilde{\boldsymbol{x}}$ and $\boldsymbol{x}$ is measured by $l_\infty$ norm in this study.

### 3.3 VIEWING TAIG FROM THE THREE PERSPECTIVES

In the TAIG attack equations i.e., equation 3 and equation 7, the integration of optimization, attention, and smoothing approaches is not obvious. This subsection explains TAIG from the perspective of optimization first, and followed by the perspectives of attention and smoothing. TAIG-S is used in the following discussion, as the discussion of TAIG-R is similar.

Using the completeness axiom, the minimization of $f$ can be written as

$$\min_{\boldsymbol{x}} f(\boldsymbol{x}) = \min_{\boldsymbol{x}} \sum_{i=1}^{N} (\text{IG}_i(\boldsymbol{x})) + f(\boldsymbol{r}). \tag{8}$$

Since $f(\boldsymbol{r})$ is independent of $\boldsymbol{x}$, it can be ignored. Taking gradient on both sides of equation 8, we have the sum of $\nabla \text{IG}_i(\boldsymbol{x})$ equal to $\nabla f(\boldsymbol{x})$, which is the same as the gradients used in PGD and FGSM for white-box attack. For ReLU networks, it can be proven that

$$\nabla f(\boldsymbol{x}) = \sum_{i=1}^{N} \nabla(\text{IG}_i(\boldsymbol{x})), \tag{9}$$

where the j-th element of $\nabla(\text{IG}_i(\boldsymbol{x}))$ is

$$\frac{\partial \text{IG}_i(\boldsymbol{x})}{\partial x_j} = \begin{cases} \frac{\partial \text{IG}_i(\boldsymbol{x})}{\partial x_j} & , \quad \text{if } i = j, \\ 0 & , \quad otherwise. \end{cases} \tag{10}$$

The proof is given in Appendix A.1. Computing the derivative of $\text{IG}_i(\boldsymbol{x})$ respect to $x_i$ and using the definition of derivative,

$$\frac{\partial \text{IG}_i(\boldsymbol{x})}{\partial x_i} = \lim_{h \to 0} \frac{\text{IG}_i(\boldsymbol{x} + h\Delta\boldsymbol{x}) - \text{IG}_i(\boldsymbol{x})}{h}, \tag{11}$$

where all the elements of $\Delta \boldsymbol{x}$ are zero, except for the i-th element being one. Using the backward difference, it can be approximated as

$$\frac{\partial \mathrm{IG}_i(\boldsymbol{x})}{\partial x_i} \approx \frac{\mathrm{IG}_i(\boldsymbol{x}) - \mathrm{IG}_i(\boldsymbol{x} - h\Delta\boldsymbol{x})}{h}, \tag{12}$$

where $h > 0$. According to the completeness axiom, if an adversarial example $\widetilde{\boldsymbol{x}}_{tar}$, whose $\mathrm{IG}_i(\widetilde{\boldsymbol{x}}_{tar}) = 0, \forall i$, then $f(\widetilde{\boldsymbol{x}}_{tar}) - f(\boldsymbol{r}) = 0$, where $\boldsymbol{r}$ is a black image in this study. In other words, the network outputs of the adversarial example and the black image are the same, which implies that the adversarial example has a high probability to be misclassified. In equation 12, $(\mathrm{IG}_i(\boldsymbol{x}) - \mathrm{IG}_i(\boldsymbol{x} - h\Delta\boldsymbol{x}))/h$ represents the slope between $\mathrm{IG}_i(\boldsymbol{x})$ and $\mathrm{IG}_i(\boldsymbol{x} - h\Delta\boldsymbol{x})$. $\mathrm{IG}_i(\boldsymbol{x})$ and $\mathrm{IG}_i(\boldsymbol{x} - h\Delta\boldsymbol{x})$ can be regarded as the integrated gradient of the i-th element of the current $\boldsymbol{x}$ and the target adversarial example. To minimize equation 8, we seek an adversarial example, whose integrated gradients are zero. Thus, the target integrated gradient i.e., $\mathrm{IG}_i(\boldsymbol{x} - h\Delta\boldsymbol{x})$ is set to zero such that it would not contribute to the network output. Setting $\mathrm{IG}_i(\boldsymbol{x} - h\Delta\boldsymbol{x})$ to zero,

$$\frac{\partial \mathrm{IG}_i(\boldsymbol{x})}{\partial x_i} \approx \frac{\mathrm{IG}_i(\boldsymbol{x})}{h}, \tag{13}$$

is obtained. Given that $h$ in equation 13 is positive and TAIG-S uses the sign of IG, we can draw the following conclusions: 1) $\mathrm{sign}(\mathrm{IG}(\boldsymbol{x}))$ can be used to approximate $\mathrm{sign}(\nabla f)$ of ReLU networks; 2) the quality of this approximation depends on equation 12, meaning that $\mathrm{sign}(\mathrm{IG}(\boldsymbol{x}))$ and $\mathrm{sign}((\nabla f))$ are not necessarily very close. To maintain the sign of $\nabla f$ for the minimization, we choose backward difference instead of forward difference in equation 12. If forward difference is used in equation 12, the term $-\alpha \times \mathrm{sign}(\mathbf{IG}(f_y, \boldsymbol{x}))$ in equation 3 would become $+\alpha \times \mathrm{sign}(\mathbf{IG}(f_y, \boldsymbol{x}))$ and TAIG-S would maximize $f$. More detailed explanation can be found in Appendix A.2.

Fig. 2 shows the distribution of normalized $\|\mathrm{sign}(\mathbf{IG}(f, \boldsymbol{x})) - \mathrm{sign}(\nabla f)\|_1$, where $\| \cdot \|_1$ is $l_1$ norm. On average, 68% elements of $\mathrm{sign}(\mathbf{IG}(f, \boldsymbol{x}))$ and $\mathrm{sign}(\nabla f)$ are the same, indicating that $\mathrm{sign}(\mathbf{IG}(f, \boldsymbol{x}))$ weakly approximates $\mathrm{sign}(\nabla f)^2$. The previous discussion is also applicable to RIG, because it also fulfills the completeness axiom. Fig. 2 also shows the distribution of normalized $\|\mathrm{sign}(\mathbf{RIG}(f, \boldsymbol{x})) - \mathrm{sign}(\nabla f)\|_1$. On average, 58% elements of $\mathrm{sign}(\mathbf{RIG}(f, \boldsymbol{x}))$ and $\mathrm{sign}(\nabla f)$ are the same. Because $\mathrm{sign}(\mathbf{IG}(f, \boldsymbol{x}))$ can weakly approximate $\mathrm{sign}(\nabla f)$, TAIG-S can be used to perform white-box attack. As for how it can enhance the transferability in black-box attacks, we believe more insights can be gained by viewing TAIG from the perspectives of attention and smoothing.

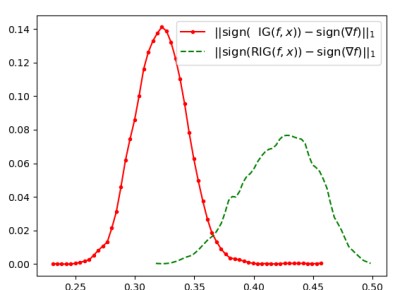

Figure 2: The relative frequency distributions of normalized $\|\mathrm{sign}(\mathbf{IG}(f, \boldsymbol{x})) - \mathrm{sign}(\nabla f)\|$ and $\|\mathrm{sign}(\mathbf{RIG}(f, \boldsymbol{x})) - \mathrm{sign}(\nabla f)\|$. The normalization is done by dividing 2 times the total number of pixels. ResNet50 is utilized to compute $\mathrm{IG}(f, \boldsymbol{x})$, $\mathrm{RIG}(f, \boldsymbol{x})$ and $\nabla f$.

Viewing from the perspective of attention, integrated gradients identify key features, which deep networks use in the prediction and allot higher integrated gradient values to them. If networks are trained to perform on the same classification task, they likely use similar key features. In equation 12, the target integrated gradient in the backward difference is set to zero. By modifying the input image through its integrated gradients, the key features are amended, and the transferability can be enhanced. To justify these arguments, Fig. 1 shows the integrated gradients of an original image from different networks and Fig. 3 shows the integrated gradients before and after TAIG-S and TAIG-R attacks. These figures indicate that 1) different models have similar integrated gradients for the same images and 2) TAIG-S and TAIG-R significantly modify the integrated gradients.

To avoid overfitting the surface of surrogate model, smoothing based on augmentation is commonly used. Equation 1 shows that TAIG-S employs intensity augmentation if the reference $\boldsymbol{r}$ is a black

---

[2]It should be highlighted that better approximation is not our goal. When $\mathrm{sign}(\mathbf{IG}(f, \boldsymbol{x}))$ is close to $\mathrm{sign}(\nabla f)$, it only implies that it is stronger for white-box attack.

image. Since $r_i = 0$, $x_i \geq 0$ and TAIG-S only uses the sign function, the term $(x_i - r_i)$ in equation 1 can be ignored. Therefore, $\text{sign}(\text{IG}(f_y, \boldsymbol{x}))$ in equation 3 only depends on $\int_{\eta=0}^{1} \frac{\partial f(\eta \boldsymbol{x})}{\partial x_i} d\eta$, whose discrete version is $\frac{1}{S} \sum_{s=0}^{S} \frac{\partial f(s \times \boldsymbol{x}/S)}{\partial x_i}$. This discrete version reveals that TAIG-S uses the sum of the gradients from intensity augmented images to perform the attack. Similarly, it can be observed from equation 1, equation 4 and equation 6 that TAIG-R applies both noise and intensity augmentation to perform the attack.

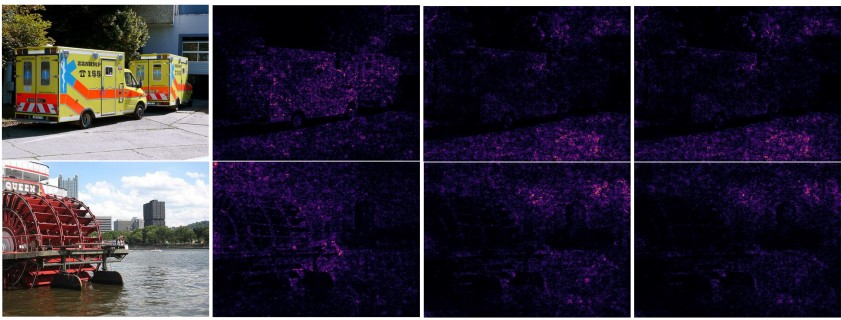

Figure 3: The first row is the original image and the IG before and after attack. The second row is the original image and the RIG before and after attack. The images from left to right are the original images, the original IG (RIG) of the image, the IG (RIG) after TAIG-S attack, and the IG (RIG) after TAIG-R attack. The results are obtained from ResNet50.

## 4 EXPERIMENTAL RESULTS

In this section, we compare the performance of the proposed TAIG-S and TAIG-R with the state-of-the-art methods, including LinBP (Guo et al., 2020), SI (Lin et al., 2020), AOA (Chen et al., 2020) and VT (Wang & He, 2021). As LinBP had shown its superiority over methods, including SGM (Wu et al., 2020a), TAP (Zhou et al., 2018) and ILA (Huang et al., 2019), they are not included in the comparisons. The comparisons are done on both undefended and advanced defense models. To verify that IG and RIG could be a good replacement of standard gradient for transferable attacks, they are also evaluated on the combinations with different methods. In the last experiment, we use different surrogate models to demonstrate that TAIG is applicable to different models and has similar performances.

### 4.1 EXPERIMENTAL SETTING

ResNet50 (He et al., 2016) is selected as a surrogate model to generate adversarial examples to compare with the state-of-the-art methods. InceptionV3 (Szegedy et al., 2016), DenseNet121 (Huang et al., 2017), MobileNetV2 (Sandler et al., 2018), SENet154 (Hu et al., 2018) and PNASNet-5-Large (Liu et al., 2018) are selected to evaluate the transferability of the adversarial examples. For the sake of convenience, the names of the networks are shortened as ResNet, Inception, DenseNet, MobileNet, SENet, and PNASNet in the rest of the paper. The networks selected for the black-box attacks were invented after or almost at the same period of ResNet, and the architecture of PNASNet was found by neural searching. The experiments are conducted on ImageNet (Russakovsky et al., 2015) validation set. 5000 images are randomly selected from images that can be correctly classified by all the networks. As with the previous works (Guo et al., 2020; Lin et al., 2020), $l_\infty$ is used to measure the difference between benign image and the corresponding adversarial example. By following LinBP, the maximum allowable perturbations are set as $\varepsilon = 0.03, 0.05, 0.1$. We also report the experimental results of $\varepsilon = 4/255, 8/255, 16/255$ in Table 6 in the appendix. The preprocessing procedure of the input varies with models. We follow the requirement to preprocess the adversarial examples before feeding them into the networks. For example, for SENet the adversarial examples are preprocessed by three steps: (1) resized to $256 \times 256$ pixels, (2) center cropped to $224 \times 224$ pixels and (3) normalized using $mean = [0.485, 0.456, 0.406]$ and $std = [0.229, 0.224, 0.225]$. Attack success rate is used as an evaluation metric to compare all the methods. Thirty sampling points are used to estimate TAIG-S. For TAIG-R, the number of turning point $E$ is set to 30 and $\tau$ is set

equal to $\varepsilon$. Because the line segment is short, each segment is estimated by one sampling point in TAIG-R. An ablation study about the number of sampling points is provided in the supplemental material Section A.6. All the experiments are performed on two NVIDIA GeForce RTX 3090 with the main code implemented using PyTorch.

Table 1: Attack success rate(%) of different methods in the untargeted setting. The symbol $*$ indicates the surrogate model used to generate the adversarial examples. The attack success rate of the surrogate model is not included in the calculation of average attack success rate. The images of AOA-SI are provided by DAmageNet with VGG19 as the surrogate model.

| Method | $\varepsilon$ | ResNet* | Inception | PNASNet | SENet | DenseNet | MobileNet | Average |
|--------|------|---------|-----------|---------|-------|----------|-----------|---------|
| AOA | 0.03 | **100.00** | 22.80 | 7.30 | 11.32 | 36.98 | 36.98 | 23.08 |
| | 0.05 | **100.00** | 36.40 | 14.64 | 22.30 | 55.00 | 47.74 | 35.22 |
| | 0.1 | **100.00** | 64.62 | 37.40 | 52.24 | 80.56 | 76.30 | 62.22 |
| SI | 0.03 | **100.00** | 21.46 | 12.68 | 15.14 | 42.58 | 43.16 | 27.00 |
| | 0.05 | **100.00** | 36.80 | 25.12 | 37.76 | 62.52 | 61.92 | 44.82 |
| | 0.1 | **100.00** | 58.40 | 48.20 | 56.68 | 91.90 | 79.74 | 66.98 |
| LinBP | 0.03 | **100.00** | 31.16 | 28.20 | **44.36** | 66.04 | 65.50 | 47.05 |
| | 0.05 | **100.00** | 62.08 | 58.88 | **77.08** | 89.84 | 88.78 | 75.33 |
| | 0.1 | **100.00** | 92.58 | 93.66 | **97.72** | 99.22 | 98.94 | 96.42 |
| TAIG-S | 0.03 | **100.00** | 24.98 | 14.06 | 20.32 | 48.66 | 50.56 | 31.72 |
| | 0.05 | **100.00** | 43.56 | 30.68 | 42.74 | 71.92 | 71.76 | 52.13 |
| | 0.1 | **100.00** | 69.68 | 60.06 | 74.78 | 91.54 | 89.70 | 77.15 |
| TAIG-R | 0.03 | **100.00** | **46.06** | **33.82** | 40.14 | **72.52** | **71.58** | **52.82** |
| | 0.05 | **100.00** | **74.22** | **66.30** | 73.46 | **93.62** | **91.56** | **79.83** |
| | 0.1 | **100.00** | **95.18** | **94.50** | 96.02 | **99.60** | **99.20** | **96.90** |

| | | | | VGG19* | | | | |
|--------|------|---------|-----------|---------|-------|----------|-----------|---------|
| Method | $\varepsilon$ | ResNet | Inception | PNASNet | SENet | DenseNet | MobileNet | Average |
| AOA-SI | 0.1 | 90.40 | 85.72 | 79.22 | 68.50 | 92.98 | 92.36 | 83.76 |
| TAIG-R | 0.1 | **98.40** | **95.95** | **95.56** | **95.84** | **95.82** | **98.40** | **97.43** |

## 4.2 Comparison with State-of-the-Arts

Usually, one method has several versions by using different back-end attacks such as FGSM, IFGSM, and MIFGSM. In the comparison, FGSM and IFGSM are selected as the back-end methods for one-step and multi-step attacks, respectively. In one-step attacks, the step size for all the methods is set to $\varepsilon$. In multi-step attacks, for LinBP, we keep the default setting where the number of iterations is 300 and the step size is 1/255 for all different $\varepsilon$. The linear backpropagation layer starts at the first residual block in the third convolution layer, which provides the best performance (Guo et al., 2020). TAIG-S and TAIG-R are run 20, 50, and 100 iterations with the same step size as LinBP for $\varepsilon = 0.03, 0.05, 0.1$, respectively. For SI, the number of scale copies is set to 5, which is the same as the default setting. The default numbers of iterations of SI and AOA are 10 and 10-20 for $\varepsilon = 16/255$ and $\varepsilon = 0.1$ respectively. We found that more iterations would improve the performance of SI and AOA. Therefore, the numbers of iterations of SI and AOA for different $\varepsilon$ are kept the same as TAIG-S and TAIG-R. The authors of AOA method provide a public dataset named DAmageNet, which consists of adversarial examples generated by combining SI and AOA with ImageNet validation set as the source images. VGG19 was taken as the surrogate model and $\varepsilon$ is set to 0.1. For comparison, TAIG-R is used to generate another set of adversarial examples with $\varepsilon = 0.1$ and VGG19 as the surrogate model. 5000 adversarial images generated by TAIG-R and 5000 images sampled from DAmageNet with the same source images are used in the evaluation. Table 1 lists the experimental results of untargeted multi-step attacks. It demonstrates that the proposed TAIG-S outperforms AOA and SI significantly but it is weaker than LinBP. The proposed TAIG-R outperforms all the state-of-the-art methods in all models, except for SENet. In terms of average attack success rate, TAIG-R achieves an average attack success rate of 52.82% under $\varepsilon = 0.03$. Comparison results of the one-step attacks are given in Table 5 in the appendix, which also shows the superiority of TAIG-R. We also provide the results of LinBP and SI under the same number of gradient calculations as TAIG-S and TAIG-R in Table 8 in the appendix. VT (Wang & He, 2021) and Admix (Wang et al., 2021), which use NIFGSM (Lin et al., 2020) and MIFGSM (Dong et al.,

2018) as the back-end methods are also compared and the experimental results are given in Table 11 in the appendix.

LinBP and SI are also examined on the target attack setting. AOA is excluded because it was not designed for target attacks. The experimental results show that TAIG-R can get an average attack success rate of 26% under $\varepsilon = 0.1$, which is the best among all the examined methods. The second best is LinBP with an average attack success rate of 10.1%. Due to limited space, the full experimental results are given in Table 7 in the appendix.

### 4.3 Evaluation on advanced defense models

In addition to the undefended models, we further examine TAIG-S and TAIG-R on six advanced defense methods. By following SI, we include the top-3 defense methods in the NIPS competition[3] and three recently proposed defense methods in our evaluation. These methods are high level representation guided denoiser (HGD, rank-1) (Liao et al., 2018), random resizing and padding (R&P, rank-2) (Xie et al., 2018), the rank-3 submission[4] (MMD), feature distillation (FD) (Liu et al., 2019), purifying perturbations via image compression model (ComDefend) (Jia et al., 2019) and random smoothing (RS) (Cohen et al., 2019)). $\varepsilon$ is set to 16/255, which is the same as the setting in the SI study (Lin et al., 2020), and ResNet is taken as the surrogate model in this evaluation. The results are listed in Table 2. The average attack success rates of TAIG-R is 70.82%, 25.61% higher than the second best – LinBP. It shows that TAIG-R outperforms all the state-of-the-art methods in attacking the advanced defense models. We also evaluate the performance of TAIG-S and TAIG-R on two advanced defense models based on adversarial image detection in Section A.5.

### 4.4 Combination with Other Methods

TAIG-S and TAIG-R can be used as back-end attacks, as IFGSM and M-IFGSM, by other methods to achieve higher transferability. Specifically, $\mathbf{IG}(x)$ and $\mathbf{RIG}(x)$ in TAIG-S and TAIG-R can replace the standard gradients in the previous methods to produce stronger attacks. In this experiment, LinBP, DI, and ILA with different back-end attacks are investigated. LinBP and ILA use the same set of back-end attacks, including IFGSM, TAIG-S, and TAIG-R. For LinBP, the back-end attacks are used to seek adversarial examples, while for ILA, the back-end attacks are used to generate directional guides. For DI, $\mathbf{IG}(x)$ and $\mathbf{RIG}(x)$ are used to replace the gradients in M-IFGSM and produce two new attacks. The two new attacks are named DI+MTAIG-S and DI+MTAIG-R. The numbers of iterations of all the back-end attacks are 20, 50, and 100 for $\varepsilon = 0.03, 0.05, 0.1$ respectively. As ILA and their back-end attacks are performed in a sequence, the number of iterations for ILA is fixed to 100, which is more effective than the default setting. And the intermediate output layer is set to be the third residual block in the second convolution layer in ResNet50 as suggested by (Guo et al., 2020). Table 4 lists the results. They indicate that TAIG-S and TAIG-R effectively enhance the transferability of other methods. As with other experiments, TAIG-R performs the best. The results of $\varepsilon = 4/255, 8/555, 16/255$, are given in Table 13 in the appendix.

Table 2: Attack success rate(%) of different methods against the advanced defense schemes. The adversarial examples are generated from ResNet.

|  | HGD | R&P | MMD | FD | Comdefend | RS | Average |
|---|---|---|---|---|---|---|---|
| AOA | 6.06 | 5.63 | 6.67 | 19.09 | 40.54 | 6.46 | 14.08 |
| SI | 26.07 | 13.03 | 16.79 | 41.37 | 66.79 | 7.59 | 28.61 |
| LinBP | 49.40 | 16.70 | 21.74 | 84.56 | 88.91 | 9.94 | 45.21 |
| TAIG-S | 32.07 | 16.04 | 20.93 | 49.89 | 81.40 | 8.69 | 34.84 |
| TAIG-R | **75.29** | **56.00** | **63.18** | **94.71** | **97.58** | **38.14** | **70.82** |

### 4.5 Evaluation on other surrogate models

To demonstrate that TAIG is also applicable to other surrogate models, we select SENet, VGG19, Inception, and DenseNet as the surrogate models and test TAIG on the same group of black-box

---

[3]https://www.kaggle.com/c/nips-2017-defense-against-adversarial-attack.
[4]https://github.com/anlthms/nips-2017/tree/master/mmd.

Table 3: Attack success rate(%) using different surrogate models. The attack success rate of the surrogate model is not included in the calculation of average success rate.

| Surrogate model | ResNet | Inception | PNASNet | SENet | DenseNet | MobileNet | Average |
|---|---|---|---|---|---|---|---|
| SENet | 90.92 | 83.18 | 84.78 | 99.66 | 91.80 | 89.70 | 88.08 |
| VGG19 | 93.96 | 84.70 | 86.20 | 86.88 | 93.64 | 96.24 | 90.27 |
| Inception | 76.36 | 100.00 | 59.30 | 59.66 | 78.04 | 80.08 | 90.69 |
| DenseNet | 99.32 | 90.26 | 89.76 | 89.64 | 99.98 | 97.54 | 93.30 |

Table 4: Attack success rate(%) of different combinations. The symbol ∗ indicates the surrogate model used to generate adversarial examples. The attack success rate of the surrogate model is not included in the calculation of average success rate.

| Method | $\varepsilon$ | ResNet ∗ | Inception | PNASNet | SENet | DenseNet | MobileNet | Average |
|---|---|---|---|---|---|---|---|---|
| ILA+ IFGSM | 0.03 | 99.98 | 30.72 | 30.62 | 42.70 | 59.76 | 61.42 | 45.04 |
| | 0.05 | **100.00** | 52.00 | 51.82 | 68.76 | 80.46 | 82.12 | 67.03 |
| | 0.1 | **100.00** | 84.12 | 85.10 | 93.96 | 96.56 | 97.04 | 91.36 |
| ILA+ TAIG-S | 0.03 | **100.00** | 40.60 | 33.20 | 51.60 | 70.84 | 72.28 | 53.70 |
| | 0.05 | **100.00** | 66.40 | 60.88 | 80.08 | 90.96 | 90.58 | 77.78 |
| | 0.1 | **100.00** | 93.62 | 93.20 | 97.96 | 99.32 | 98.74 | 96.57 |
| ILA+ TAIG-R | 0.03 | **100.00** | **49.24** | **41.50** | **57.56** | **78.68** | **78.04** | **61.00** |
| | 0.05 | **100.00** | **74.76** | **71.64** | **85.18** | **94.90** | **93.42** | **83.98** |
| | 0.1 | **100.00** | **96.54** | **96.48** | **98.58** | **99.72** | **99.44** | **98.15** |
| LinBP+ IFGSM | 0.03 | 99.98 | 28.36 | 25.00 | 36.56 | 61.96 | 61.84 | 42.74 |
| | 0.05 | **100.00** | 57.34 | 55.44 | 71.50 | 87.06 | 86.52 | 71.57 |
| | 0.1 | **100.00** | 88.82 | 90.62 | 95.76 | 98.86 | 98.44 | 94.50 |
| LinBP+ TAIG-S | 0.03 | **99.98** | 48.26 | 34.94 | 47.98 | 79.42 | 77.62 | 57.64 |
| | 0.05 | **100.00** | 81.24 | 73.88 | 84.48 | 87.08 | 95.54 | 84.44 |
| | 0.1 | **100.00** | 98.90 | 98.56 | 99.32 | 99.90 | 99.72 | 99.28 |
| LinBP+ TAIG-R | 0.03 | **99.98** | **60.14** | **47.70** | **59.26** | **85.52** | **84.70** | **67.46** |
| | 0.05 | **100.00** | **88.80** | **85.80** | **91.18** | **98.52** | **97.58** | **92.38** |
| | 0.1 | **100.00** | **99.34** | **99.24** | **99.62** | **99.92** | **99.86** | **99.60** |
| DI+M IFGSM | 0.03 | 99.98 | 19.34 | 16.12 | 20.02 | 42.70 | 40.71 | 27.78 |
| | 0.05 | **100.00** | 31.54 | 28.50 | 35.94 | 58.94 | 57.42 | 42.47 |
| | 0.1 | **100.00** | 57.64 | 57.22 | 67.36 | 83.66 | 84.94 | 70.16 |
| DI+M TAIG-S | 0.03 | **100.00** | 39.50 | 25.94 | 33.14 | 64.16 | 64.02 | 45.35 |
| | 0.05 | **100.00** | 59.30 | 47.04 | 59.08 | 83.88 | 82.60 | 66.38 |
| | 0.1 | **100.00** | 86.16 | 81.10 | 86.82 | 97.22 | 96.16 | 89.49 |
| DI+M TAIG-R | 0.03 | **100.00** | **58.46** | **46.68** | **51.72** | **81.46** | **78.62** | **63.39** |
| | 0.05 | **100.00** | **82.80** | **75.94** | **80.46** | **96.20** | **93.96** | **85.87** |
| | 0.1 | **100.00** | **97.86** | **97.32** | **97.84** | **99.76** | **99.54** | **98.46** |

models. Because TAIG-S and TAIG-R are almost the same, except for the paths of computing integrated gradients and TAIG-R outperforms TAIG-S in the previous experiments, we select TAIG-R as an example. In this experiment, we set $\varepsilon$ to 16/255, and the results are reported in Table 3. It indicates that TAIG-R performs similarly on different surrogate models.

## 5 CONCLUSION

In this paper, we propose a new attack algorithm named Transferable Attack based on Integrated Gradients. It tightly integrates three common approaches in crafting adversarial examples to generate highly transferable adversarial examples. Two versions of the algorithm, one based on straight-line integral, TAIG-S, and the other based on random piecewise linear path integral, TAIG-R, are studied. Extensive experiments, including attacks on undefended and defended models, untargeted and targeted attacks and combinations with the previous methods, are conducted. The experimental results demonstrate the effectiveness of the proposed algorithm. In particular, TAIG-R outperforms all the state-of-the-art methods in all the settings.

## ACKNOWLEDGMENT

This work is partially supported by the Ministry of Education, Singapore through Academic Research Fund Tier 1, RG73/21.

## ETHICS STATEMENT

Hereby, we assure that the experimental results are reported accurately and honestly. The experimental settings including data splits, hyperparameters, and GPU resources used are clearly introduced in Section 4 and the source code will be shared after this paper is published. The ImageNet dataset and source code from AOA and LinBP attack are used in this paper and they are cited in Section 4.

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

## A    APPENDIX

### A.1    THE PROOF OF EQUATION 10

For ReLU networks[5], it can be proven that the j-th element of $\nabla(\text{IG}_i(\boldsymbol{x}))$ is:

$$\frac{\partial \text{IG}_i(\boldsymbol{x})}{\partial x_j} = \begin{cases} \frac{\partial \text{IG}_i(\boldsymbol{x})}{\partial x_j} & , \quad \text{if } i = j, \\ \\ 0 & , \quad \textit{otherwise.} \end{cases}$$

Considering $\frac{\partial \text{IG}_i(\boldsymbol{x})}{\partial x_j} = \frac{\partial}{\partial x_j}\left\{(x_i - r_i) \times \int_{\eta=0}^{1} \frac{\partial f(\boldsymbol{r} + \eta \times (\boldsymbol{x} - \boldsymbol{r}))}{\partial x_i}\mathrm{d}\eta\right\}$. Using the product rule, we have

$$\frac{\partial(x_i - r_i)}{\partial x_j} \times \int_{\eta=0}^{1} \frac{\partial f(\boldsymbol{r} + \eta \times (\boldsymbol{x} - \boldsymbol{r}))}{\partial x_i}\mathrm{d}\eta + (x_i - r_i) \times \int_{\eta=0}^{1} \frac{\partial}{\partial x_j}\frac{\partial f(\boldsymbol{r} + \eta \times (\boldsymbol{x} - \boldsymbol{r}))}{\partial x_i}\mathrm{d}\eta$$

$\frac{\partial(x_i - r_i)}{\partial x_j} = 0$, if $i \neq j$; otherwise, $\frac{\partial(x_i - r_i)}{\partial x_j} = 1$. Thus, the first term becomes

$$\frac{\partial(x_i - r_i)}{\partial x_j} \times \int_{\eta=0}^{1} \frac{\partial f(\boldsymbol{r} + \eta \times (\boldsymbol{x} - \boldsymbol{r}))}{\partial x_i}\mathrm{d}\eta = \begin{cases} \int_{\eta=0}^{1} \frac{\partial f(\boldsymbol{r} + \eta \times (\boldsymbol{x} - \boldsymbol{r}))}{\partial x_i}\mathrm{d}\eta & , \quad \text{if } i = j, \\ \\ 0 & , \quad \textit{otherwise.} \end{cases}$$

$\int_{\eta=0}^{1} \frac{\partial}{\partial x_j}\frac{\partial f(\boldsymbol{r} + \eta \times (\boldsymbol{x} - \boldsymbol{r}))}{\partial x_i}\mathrm{d}\eta$ in the second term is zero because of the ReLU functions in the network. Thus,

$$\frac{\partial \text{IG}_i(\boldsymbol{x})}{\partial x_j} = \begin{cases} \int_{\eta=0}^{1} \frac{\partial f(\boldsymbol{r} + \eta \times (\boldsymbol{x} - \boldsymbol{r}))}{\partial x_i}\mathrm{d}\eta & , \quad \text{if } i = j, \\ \\ 0 & , \quad \textit{otherwise.} \end{cases} \tag{14}$$

Since $\frac{\partial \text{IG}_i(\boldsymbol{x})}{\partial x_i} = \int_{\eta=0}^{1} \frac{\partial f(\boldsymbol{r} + \eta \times (\boldsymbol{x} - \boldsymbol{r}))}{\partial x_i}\mathrm{d}\eta$, equation 14 can be written as

$$\frac{\partial \text{IG}_i(\boldsymbol{x})}{\partial x_j} = \begin{cases} \frac{\partial \text{IG}_i(\boldsymbol{x})}{\partial x_j} & , \quad \text{if } i = j, \\ \\ 0 & , \quad \textit{otherwise.} \end{cases} \tag{15}$$

### A.2    WHY USE BACKWARD DIFFERENCE

Some may question why backward difference is used in equation 12, instead of forward difference i.e., $(\text{IG}_i(\boldsymbol{x} + h\Delta\boldsymbol{x}) - \text{IG}_i(\boldsymbol{x}))/h$. If forward difference is used and the target integrated gradient, i.e., $\text{IG}_i(\boldsymbol{x} + h\Delta\boldsymbol{x})$ in forward difference is set to zero,

$$\frac{\partial \text{IG}_i(\boldsymbol{x})}{\partial x_i} \approx -\frac{\text{IG}_i(\boldsymbol{x})}{h} \tag{16}$$

where $h > 0$. The attack equation would become

$$\widetilde{\boldsymbol{x}} = \boldsymbol{x} + \alpha \times \text{sign}(\mathbf{IG}(f_y, \boldsymbol{x})). \tag{17}$$

Considering the discrete vector form of $\mathbf{IG}$,

$$\mathbf{IG}(f, \boldsymbol{x}, \boldsymbol{r}) = (\boldsymbol{x} - \boldsymbol{r}) \circ \frac{1}{m}\sum_{i=0}^{m} \nabla f(\boldsymbol{r} + \frac{i}{m} \times (\boldsymbol{x} - \boldsymbol{r})), \tag{18}$$

---

[5]The term ReLU network is to refer to the networks with second-order derivatives being zero (Wang et al., 2020) due to their computational unit, $ReLU(\boldsymbol{w}^T\boldsymbol{v})$, where $\boldsymbol{w}$ is a network parameter and $\boldsymbol{v}$ is the input from a previous layer.

where $\circ$ is the Hadamard product also known as element-wise product, it is noted that when $i/m$, which is the discrete version of $\eta$ in equation 1, is close to one, $\nabla f(\boldsymbol{r}+\frac{i}{m}\times(\boldsymbol{x}-\boldsymbol{r}))$ can approximate $\nabla f(\boldsymbol{x})$. If we only use $\nabla f(\boldsymbol{r} + \frac{i}{m} \times (\boldsymbol{x} - \boldsymbol{r}))$, whose $i/m$ is close to one to compute equation 17, equation 17 is an approximation of gradient ascent. On the contrary, if we only use $\nabla f(\boldsymbol{r}+\frac{i}{m}\times(\boldsymbol{x}-\boldsymbol{r}))$, whose $i/m$ is close to one to compute the integrated gradients in the proposed attack equation i.e., equation 3, equation 3 is in fact an approximation of gradient decent. Since we would like to minimize $f$, backward difference is applied to equation 12 and TAIG-S in equation 3 performs an approximation of gradient decent.

## A.3 COMPARISON WITH STATE-OF-THE-ARTS

### A.3.1 MORE RESULT ON THE MAIN COMPARISONS

In this section, we provide more experimental results about the comparisons on LinBP, SI and AOA omitted in the main paper. The experimental results of TAIG-S and TAIG-R for untargeted attack under $\varepsilon = 4/255, 8/255, 16/255$ running with 20, 50 and 100 iterations are given in Table 6. The results of different methods using FGSM as the back-end method for one-step attacks are summarized in Table 5. Table 7 lists the experimental results of different methods on the target attack setting. These results highlight the effectiveness of the proposed methods, in particular, TAIG-R.

The comparison experiments between LinBP, SI, TAIG-S, and TAIG-R are also conducted under different $\varepsilon$ with the same number of gradient calculations, and the results are listed in Table 8. For AOA, the calculation of SGLPR is too slow, and we find that the average attack success rate of AOA only changes from 32.07 (100 iterations) to 32.19 (300 iterations) under $\varepsilon = 16/255$. As the result of AOA is not as good as LinBP and SI, we do not run it for more iterations. Table 8 shows that more number of gradient calculations will slightly improve the performance of SI and LinBP, but they are still not as good as TAIG-R.

Table 5: Attack success rate(%) of different methods using FGSM for one-step attack in the untargeted setting. The symbol $*$ indicates the surrogate model for generating the adversarial examples.

| Method | $\epsilon$ | ResNet* | Inception | PNASNet | SENet | DenseNet | MobilenNet | Average |
|---|---|---|---|---|---|---|---|---|
| AOA | 0.03 | 90.62 | 25.78 | 9.60 | 11.10 | 28.92 | 28.10 | 20.70 |
| | 0.05 | 88.72 | 38.44 | 16.20 | 20.70 | 39.90 | 40.20 | 31.09 |
| | 0.1 | 88.94 | 62.40 | 33.00 | 44.00 | 60.34 | 72.10 | 54.37 |
| SI | 0.03 | 61.26 | 20.00 | 10.18 | 10.54 | 20.10 | 32.86 | 18.74 |
| | 0.05 | 77.68 | 37.26 | 23.28 | 24.78 | 49.48 | 54.80 | 37.92 |
| | 0.1 | 91.90 | 66.10 | 53.50 | 56.82 | 78.66 | 83.96 | 67.81 |
| LinBP | 0.03 | 74.46 | 16.06 | 8.02 | 12.08 | 29.82 | 34.12 | 20.02 |
| | 0.05 | 75.58 | 25.46 | 25.62 | 22.56 | 42.66 | 50.42 | 33.34 |
| | 0.1 | 84.84 | 45.12 | 33.00 | 45.50 | 64.46 | 81.44 | 53.90 |
| TAIG-S | 0.03 | 84.00 | 25.32 | 14.96 | 17.48 | 38.80 | 39.30 | 27.17 |
| | 0.05 | 87.16 | 38.56 | 27.24 | 31.16 | 53.70 | 56.30 | 41.39 |
| | 0.1 | 91.52 | 61.32 | 51.30 | 57.02 | 76.46 | 83.28 | 65.88 |
| TAIG-R | 0.03 | **92.92** | **34.32** | **24.14** | **24.10** | **49.12** | **48.48** | **36.03** |
| | 0.05 | **96.48** | **53.64** | **44.02** | **46.36** | **68.06** | **66.86** | **55.79** |
| | 0.1 | **98.94** | **79.16** | **75.98** | **76.48** | **89.22** | **90.24** | **82.22** |

### A.3.2 COMPARISON WITH LINBP ON VGG19

Section 4.2 shows that LinBP performs the second-best among all the methods. Therefore, we compare TAIG-R with LinBP using VGG19 as a surrogate model under $\varepsilon = 16/255$. LinBP is sensitive to the choice of the position from where the network is modified. To find the best position to start the model modification, we test all the possible position in VGG19 separately and generate adversarial examples from these 16 different positions with 100 iterations. Then we use the six black-box models to compare their performance. Table 9 gives the experimental results. We select SENet and ResNet as two examples and show how the attack success rate of LinBP varies with the choice of the position in VGG-19 in Fig. 4. Based on Table 9, we select the two modified positions with the

Table 6: Attack success rate(%) of different methods in the untargeted setting. The symbol ∗ indicates the surrogate model for generating the adversarial examples.

| Method | $\varepsilon$ | ResNet* | Inception | PNASNet | SENet | DenseNet | MobileNet | Average |
|--------|------|---------|-----------|---------|-------|----------|-----------|---------|
| AOA | 4/255 | 99.80 | 10.64 | 1.50 | 2.64 | 11.18 | 12.34 | 7.66 |
| | 8/255 | 99.86 | 20.50 | 4.44 | 6.98 | 22.36 | 24.16 | 15.69 |
| | 16/255 | 99.78 | 40.70 | 11.50 | 18.92 | 40.78 | 48.44 | 32.07 |
| SI | 4/255 | 99.64 | 8.84 | 3.88 | 5.42 | 19.10 | 20.50 | 11.55 |
| | 8/255 | **100.00** | 20.82 | 11.76 | 15.90 | 41.12 | 42.74 | 26.47 |
| | 16/255 | **100.00** | 44.82 | 32.06 | 42.06 | 71.72 | 69.82 | 52.10 |
| LinBP | 4/255 | **99.98** | 8.78 | 6.60 | **12.20** | 25.04 | 27.32 | 15.99 |
| | 8/255 | **100.00** | 29.98 | 27.26 | **43.40** | 64.88 | 64.68 | 46.04 |
| | 16/255 | **100.00** | 74.30 | 73.98 | **88.50** | 94.78 | 94.00 | 85.11 |
| TAIG-S | 4/255 | **99.98** | 10.10 | 4.54 | 6.60 | 21.62 | 25.02 | 13.58 |
| | 8/255 | **100.00** | 26.14 | 15.02 | 22.64 | 49.80 | 52.74 | 33.27 |
| | 16/255 | **100.00** | 55.34 | 43.04 | 57.86 | 82.08 | 80.66 | 63.80 |
| TAIG-R | 4/255 | 99.96 | **17.04** | **8.66** | 11.60 | **35.38** | **37.28** | **21.99** |
| | 8/255 | **100.00** | **45.44** | **34.00** | 41.62 | **73.72** | **73.86** | **53.73** |
| | 16/255 | **100.00** | **84.30** | **78.82** | 84.72 | **97.00** | **95.90** | **88.15** |

Table 7: Attack success rate(%) of different methods using IFGSM in the targeted setting. The symbol ∗ indicates the surrogate model for generating the adversarial examples.

| Method | $\epsilon$ | ResNet* | Inception | PNASNet | SENet | DenseNet | MobilenNet | Average |
|--------|------|---------|-----------|---------|-------|----------|-----------|---------|
| SI | 0.03 | **99.36** | 0.00 | 0.00 | 0.00 | 0.04 | 0.08 | 0.02 |
| | 0.05 | **99.98** | 0.14 | 0.12 | 0.14 | 0.70 | 0.28 | 0.28 |
| | 0.10 | **100.00** | 0.48 | 0.00 | 0.98 | 2.60 | 1.02 | 1.02 |
| LinBP | 0.03 | 99.00 | 0.08 | 0.34 | 0.78 | 1.56 | 0.76 | 0.70 |
| | 0.05 | 99.58 | 0.94 | 2.00 | 3.58 | 6.90 | 3.56 | 3.40 |
| | 0.10 | 99.48 | 5.58 | 8.22 | 9.18 | 18.58 | 8.96 | 10.10 |
| TAIG-S | 0.03 | 98.06 | 0.02 | 0.04 | 0.12 | 0.76 | 0.34 | 0.26 |
| | 0.05 | 98.28 | 0.42 | 0.66 | 0.98 | 3.46 | 1.76 | 1.46 |
| | 0.10 | 98.36 | 1.86 | 3.52 | 4.74 | 11.30 | 4.34 | 5.15 |
| TAIG-R | 0.03 | 98.06 | **0.62** | **1.18** | **1.16** | **4.52** | **2.58** | **2.01** |
| | 0.05 | 98.32 | **4.66** | **7.26** | **7.08** | **18.54** | **9.60** | **9.43** |
| | 0.10 | 98.04 | **17.00** | **31.12** | **20.48** | **40.98** | **20.56** | **26.03** |

best performance and use them to generate adversarial examples with 3000 gradient calculations. The results are listed in Table 10.

### A.3.3 COMPARISON WITH VT AND ADMIX

In this section, we compare TAIG-R with Admix (Wang et al., 2021) and VT (Wang & He, 2021), which use other back-end methods. Admix uses MIFGSM as the basic iteration method (MI-Admix) and VT has two different versions NI-VT and MI-VT, which use MIFGSM and NIFGSM as their back-end methods respectively. VT and Admix both use InceptionV3 as the surrogate model in their study and they also provided 1000 images used in their evaluation. $\varepsilon$ was set to 16/255 in their experiment. Thus, we run TAIG-R on InceptionV3 on the same 1000 images they used under the same value of $\varepsilon$. All the methods are run with 3000 gradient calculations. Table 11 lists the attack success rate of these methods on different models and shows that TAIG-R outperforms MI-Admix, MI-VT, and NI-VT in all the models. NI-VT is slightly better than MI-VT. As NI-VT performs the best, we further run NI-VT on the 5000 images used in our evaluation with ResNet50 as a surrogate model under different $\varepsilon$ and the results are listed in Table 12.

### A.4 COMBINATIONS WITH OTHER METHODS

In Section 4.3, LinBP, DI, and ILA with different back-end attacks are investigated under $\varepsilon = 0.03, 0.05, 0.1$. In this section, we provide the experimental results of the same setting under

Table 8: Attack success rate(%) of different methods under the same number of gradient calculations with TAIG-S and TAIG-R using IFGSM in the untargeted setting. The symbol ∗ indicates the surrogate model for generating the adversarial examples.

| Methods | $\varepsilon$ | Number of gradient calculations | ResNet* | Inception | Pnasnet | Senet | Densenet | Mobilenet | Average |
|---|---|---|---|---|---|---|---|---|---|
| SI | 4/255 | 600 | **99.90** | 8.12 | 3.58 | 4.90 | 17.34 | 19.42 | 10.67 |
| | 8/255 | 1500 | 100.00 | 20.40 | 11.12 | 16.12 | 40.04 | 42.54 | 26.04 |
| | 16/255 | 3000 | 100.00 | 48.48 | 35.68 | 48.96 | 73.92 | 72.84 | 55.98 |
| | 0.03 | 600 | 100.00 | 22.28 | 12.06 | 17.80 | 42.28 | 44.38 | 27.76 |
| | 0.05 | 1500 | 100.00 | 39.40 | 26.68 | 37.48 | 64.78 | 65.10 | 46.69 |
| | 0.1 | 3000 | 100.00 | 68.68 | 61.26 | 72.56 | 88.66 | 87.56 | 75.74 |
| LinBP | 4/255 | 600 | 99.98 | 8.70 | 6.30 | **11.70** | 24.72 | 26.62 | 15.61 |
| | 8/255 | 1500 | 100.00 | 29.52 | 27.10 | **43.16** | 64.12 | 63.22 | 45.42 |
| | 16/255 | 3000 | 100.00 | 74.02 | 74.50 | **87.48** | 94.86 | 94.68 | 85.11 |
| | 0.03 | 600 | 100.00 | 30.48 | 27.98 | **43.78** | 65.50 | 64.52 | 46.45 |
| | 0.05 | 1500 | 100.00 | 61.80 | 59.40 | **77.80** | 89.28 | 88.78 | 75.41 |
| | 0.1 | 3000 | 100.00 | 93.62 | 94.22 | **98.02** | 99.24 | 98.88 | 96.80 |
| TAIG-S | 4/255 | 600 | 99.98 | 10.10 | 4.54 | 6.60 | 21.62 | 25.02 | 13.58 |
| | 8/255 | 1500 | **100.00** | 26.14 | 15.02 | 22.64 | 49.80 | 52.74 | 33.27 |
| | 16/255 | 3000 | **100.00** | 55.34 | 43.04 | 57.86 | 82.08 | 80.66 | 63.80 |
| | 0.03 | 600 | **100.00** | 24.98 | 14.06 | 20.32 | 48.66 | 50.56 | 31.72 |
| | 0.05 | 1500 | **100.00** | 43.56 | 30.68 | 42.74 | 71.92 | 71.76 | 52.13 |
| | 0.1 | 3000 | **100.00** | 69.68 | 60.06 | 74.78 | 91.54 | 89.70 | 77.15 |
| TAIG-R | 4/255 | 600 | 99.96 | **17.04** | **8.66** | 11.60 | **35.38** | **37.28** | **21.99** |
| | 8/255 | 1500 | **100.00** | **45.44** | **34.00** | 41.62 | **73.72** | **73.86** | **53.73** |
| | 16/255 | 3000 | **100.00** | **84.30** | **78.82** | 84.72 | **97.00** | **95.90** | **88.15** |
| | 0.03 | 600 | **100.00** | **46.06** | **33.82** | 40.14 | **72.52** | **71.58** | **52.82** |
| | 0.05 | 1500 | **100.00** | **74.22** | **66.30** | 73.46 | **93.62** | **91.56** | **79.83** |
| | 0.1 | 3000 | **100.00** | **95.18** | **94.50** | 96.02 | **99.60** | **99.20** | **96.90** |

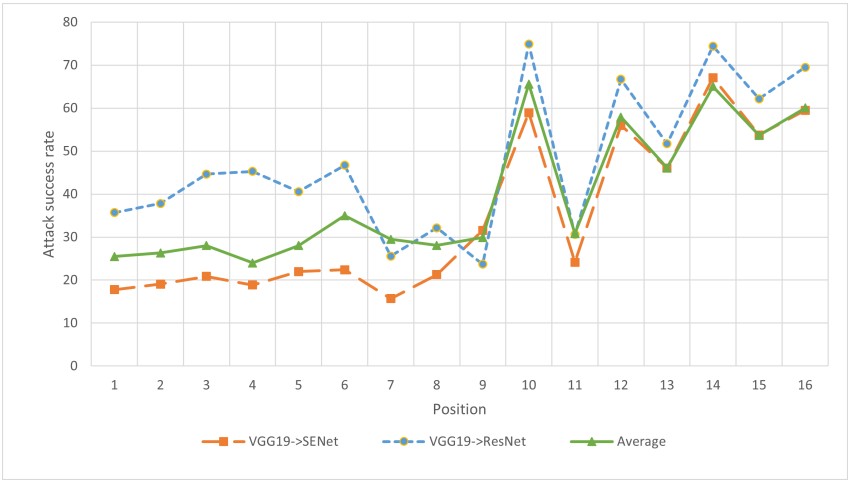

Figure 4: An illustration of how the attack success rate of LinBP varies with the choice of the position in VGG19.

$\varepsilon = 4/255, 8/255, 16/255$. The results are given in Table 13. It demonstrates that TAIG-R and TAIG-S improve the transferability of other methods. Besides, the same combinations are also examined in the target setting and the experimental results are listed in Table 14. It demonstrates that TAIG-S and TAIG-R also improve the transferability of different methods in target attacks. However, the combinations of TAIG-S and TAIG-R with other methods do not consistently improve the performance of TAIG-S and TAIG-R, which is different from the observations in untargeted attacks. Through the comparison of Table 7 and Table 14, it is noted that the transferability of TAIG-R would

Table 9: Attack success rate(%) of different modification position of LinBP in the untargeted setting. The surrogate model is VGG19 and the attack success rate is computed from 100 gradient calculations.

| LinBP Poistion | ResNet | Inception | PNASNet | SENet | DenseNet | MobileNet | Average |
|---|---|---|---|---|---|---|---|
| 1 | 35.71 | 20.67 | 9.19 | 17.76 | 24.52 | 47.48 | 25.89 |
| 2 | 37.81 | 19.76 | 10.19 | 19.05 | 25.92 | 49.38 | 27.02 |
| 3 | 44.67 | 20.38 | 10.10 | 20.86 | 27.05 | 54.57 | 29.61 |
| 4 | 45.29 | 14.14 | 7.00 | 18.86 | 23.05 | 51.81 | 26.69 |
| 5 | 40.57 | 18.33 | 10.57 | 22.00 | 27.95 | 55.24 | 29.11 |
| 6 | 46.71 | 24.52 | 11.19 | 22.38 | 38.71 | 65.62 | 34.86 |
| 7 | 25.57 | 22.24 | 13.29 | 15.71 | 28.19 | 54.21 | 26.54 |
| 8 | 32.14 | 19.05 | 11.33 | 21.29 | 28.05 | 53.81 | 27.61 |
| 9 | 23.71 | 16.86 | 20.05 | 31.57 | 54.76 | 27.95 | 29.15 |
| 10 | 74.90 | 56.10 | 53.48 | 58.90 | 70.48 | 82.29 | 66.03 |
| 11 | 30.62 | 23.10 | 16.48 | 24.14 | 32.19 | 51.43 | 29.66 |
| 12 | 66.71 | 47.19 | 45.57 | 56.00 | 63.76 | 75.24 | 59.08 |
| 13 | 51.71 | 35.24 | 32.00 | 46.10 | 51.57 | 65.52 | 47.02 |
| 14 | 74.43 | 49.10 | 58.57 | 67.10 | 71.38 | 81.10 | 66.95 |
| 15 | 62.19 | 37.81 | 43.33 | 53.76 | 59.00 | 74.52 | 55.10 |
| 16 | 69.52 | 43.57 | 53.00 | 59.52 | 66.10 | 77.52 | 61.54 |

Table 10: Attack success rate(%) of LinBP using the best two modified VGG19 as a surrogate model. The attack success rate is computed from 3000 gradient calculations.

| LinBP Poistion | ResNet | Inception | PNASNet | SENet | DenseNet | MobileNet | Average |
|---|---|---|---|---|---|---|---|
| 10 | 83.16 | 67.70 | 64.76 | 67.68 | 80.74 | 88.12 | 75.36 |
| 14 | 79.32 | 51.58 | 59.80 | 73.42 | 74.32 | 85.70 | 70.69 |
| TAIG-R | **93.96** | **84.70** | **86.20** | **86.88** | **93.64** | **96.24** | **90.27** |

be harmed when combined with other methods in the target setting. And the performance of TAIG-S will decrease when combined with ILA in the target setting.

## A.5 EVALUATION ON ADVANCED DEFENSE MODEL BASED ON DETECTION

In this section, we evaluate the performance of TAIG-S and TAIG-R on two advanced defense models based on adversarial image detection. These two methods are Feature squeezing (FS (Xu et al., 2018)) and Turning a Weakness into a Strength (TWS Hu et al. (2019)). We use their default settings and examine their detection rate on the different methods. For each of the methods, 5000 adversarial images generated under $\varepsilon = 16/255$ with 3000 gradient calculations are used in this evaluation. The detection rate and the attack success rate after detection (ASRD) are listed in Table 15. The ASRD is defined as the number of successful attacks over 5000. The detection rates of both detectors on AOA are low, but the ASRD of AOA is the lowest one. Table 15 shows that TAIG-R has the highest ASRD among all the methods.

Table 11: success rate(%) of VT and Admix using MIFGSM and NIFGSM in the untargeted setting with $\varepsilon = 16/255$. The symbol $*$ indicates the surrogate model for generating the adversarial examples.

| Method | Inception* | PNASNet | SENet | DenseNet | MobilenNet | ResNet | Average |
|---|---|---|---|---|---|---|---|
| MI-Admix | 99.10 | 51.80 | 55.10 | 67.80 | 69.60 | 67.70 | 62.40 |
| MI-VT | 98.80 | 60.40 | 60.40 | 68.50 | 70.90 | 69.60 | 65.96 |
| NI-VT | 98.90 | 61.40 | 62.10 | 68.60 | 71.80 | 70.70 | 66.92 |
| TAIG-R | **100.00** | **65.30** | **64.60** | **79.90** | **82.90** | **79.80** | **74.50** |

Table 12: Attack success rate(%) of NI-VT under the same number of gradient calculations with TAIG-S and TAIG-R in the untargeted setting. The symbol $*$ indicates the surrogate model for generating the adversarial examples.

| Methods | $\varepsilon$ | Number of gradient calculations | ResNet$^*$ | Inception | Pnasnet | Senet | Densenet | Mobilenet | Average |
|---|---|---|---|---|---|---|---|---|---|
| NI-VT | 4/255 | 600 | 94.40 | 14.06 | 6.16 | 5.88 | 16.12 | 23.54 | 13.15 |
| | 8/255 | 1500 | 95.20 | 31.82 | 18.28 | 18.50 | 37.58 | 42.00 | 29.64 |
| | 16/255 | 3000 | 96.20 | 61.22 | 46.56 | 47.78 | 65.16 | 70.10 | 58.16 |
| | 0.03 | 600 | 95.40 | 32.94 | 18.62 | 18.90 | 38.40 | 42.68 | 30.31 |
| | 0.05 | 1500 | 95.80 | 52.70 | 36.16 | 34.86 | 56.94 | 61.14 | 48.36 |
| | 0.1 | 3000 | 96.70 | 79.88 | 71.42 | 68.23 | 83.20 | 84.66 | 77.48 |
| TAIG-R | 4/255 | 600 | **99.96** | **17.04** | **8.66** | **11.60** | **35.38** | **37.28** | **21.99** |
| | 8/255 | 1500 | **100.00** | **45.44** | **34.00** | **41.62** | **73.72** | **73.86** | **53.73** |
| | 16/255 | 3000 | **100.00** | **84.30** | **78.82** | **84.72** | **97.00** | **95.90** | **88.15** |
| | 0.03 | 600 | **100.00** | **46.06** | **33.82** | **40.14** | **72.52** | **71.58** | **52.82** |
| | 0.05 | 1500 | **100.00** | **74.22** | **66.30** | **73.46** | **93.62** | **91.56** | **79.83** |
| | 0.1 | 3000 | **100.00** | **95.18** | **94.50** | **96.02** | **99.60** | **99.20** | **96.90** |

Table 13: Attack success rate(%) of different combinations in the untargeted setting. The symbol $*$ indicates the surrogate model for generating adversarial examples.

| Method | $\epsilon$ | ResNet$^*$ | Inception | Pnasnet | Senet | Densenet | Mobilenet | Average |
|---|---|---|---|---|---|---|---|---|
| ILA+ IFGSM | 4/255 | **99.98** | 10.28 | 7.96 | 12.64 | 24.68 | 27.82 | 16.68 |
| | 8/255 | **100.00** | 27.16 | 26.08 | 38.94 | 55.84 | 58.42 | 41.29 |
| | 16/255 | 99.96 | 61.74 | 61.80 | 78.72 | 86.66 | 88.24 | 75.43 |
| ILA+ TAIG-S | 4/255 | 99.94 | 11.96 | 6.78 | 13.32 | 27.84 | 32.26 | 18.43 |
| | 8/255 | **100.00** | 36.00 | 28.62 | 47.98 | 68.98 | 69.52 | 50.22 |
| | 16/255 | **100.00** | 75.98 | 72.76 | 88.62 | 95.06 | 95.18 | 85.52 |
| ILA+ TAIG-R | 4/255 | 99.96 | **14.66** | **9.06** | **15.68** | **33.74** | **36.54** | **21.94** |
| | 8/255 | **100.00** | **42.56** | **35.10** | **52.14** | **73.82** | **74.68** | **55.66** |
| | 16/255 | **100.00** | **83.10** | **80.62** | **90.98** | **97.18** | **96.90** | **89.76** |
| LinBP+ IFGSM | 4/255 | 99.96 | 10.08 | 6.82 | 11.76 | 27.50 | 29.42 | 17.12 |
| | 8/255 | **100.00** | 30.48 | 27.82 | 41.66 | 64.90 | 65.14 | 46.00 |
| | 16/255 | **100.00** | 72.38 | 72.50 | 85.28 | 94.12 | 93.88 | 83.63 |
| Linbp+ TAIG-S | 4/255 | 99.66 | 14.30 | 7.32 | 12.32 | 33.88 | 36.96 | 20.96 |
| | 8/255 | 99.9 | 50.66 | 37.14 | 51.12 | 80.40 | 79.62 | 59.79 |
| | 16/255 | 99.98 | 91.58 | 87.92 | 93.24 | 99.10 | 98.44 | 94.06 |
| Linbp+ TAIG-R | 4/255 | 99.70 | **21.24** | **12.52** | **18.10** | **43.12** | **46.46** | **28.29** |
| | 8/255 | 99.98 | **62.94** | **50.96** | **63.72** | **87.96** | **86.36** | **70.39** |
| | 16/255 | 99.98 | **95.52** | **94.20** | **96.78** | **99.58** | **99.08** | **97.03** |
| DI+ MIFGSM | 4/255 | 99.94 | 8.56 | 5.80 | 7.08 | 19.80 | 20.98 | 12.44 |
| | 8/255 | **100.00** | 18.55 | 15.20 | 20.32 | 41.72 | 41.08 | 27.37 |
| | 16/255 | **100.00** | 39.32 | 38.12 | 47.74 | 70.06 | 68.00 | 52.65 |
| DI+ MTAIG-S | 4/255 | **99.98** | 14.22 | 6.74 | 9.64 | 29.44 | 31.62 | 18.33 |
| | 8/255 | **100.00** | 35.76 | 22.60 | 31.56 | 61.40 | 62.00 | 42.66 |
| | 16/255 | **100.00** | 68.32 | 57.68 | 69.16 | 89.44 | 88.24 | 74.57 |
| DI+ MTAIG-R | 4/255 | 99.96 | **21.32** | **12.26** | **15.30** | **40.62** | **43.38** | **26.58** |
| | 8/255 | **100.00** | **55.48** | **42.88** | **48.92** | **80.12** | **78.54** | **61.19** |
| | 16/255 | **100.00** | **89.50** | **85.72** | **89.16** | **98.00** | **97.26** | **91.93** |

## A.6 ABLATION STUDY

In this section, we investigate the influence of the number of sampling points. In this experiment, $\varepsilon$ is set to 8/255 and 1000 images are sampled from the 5000 images used in the evaluation. (Sundararajan et al., 2017) pointed out that the sampling points between 20 to 300 are enough to approximate the integral. Following Sundararajan et al.'s suggestion, the number of sampling points $S$ for TAIG-S starts from 20 with an increment of 10 in each of the tests. We find that with the increase of the

Table 14: Attack success rate(%) of different combinations in the target setting. The symbol $*$ indicates the surrogate model for generating adversarial examples.

| Method | $\varepsilon$ | ResNet$^*$ | Inception | PNASNet | SENet | DenseNet | MobilenNet | Average |
|---|---|---|---|---|---|---|---|---|
| ILA+ IFGSM | 0.03 | 58.86 | 0.10 | 0.10 | 0.26 | 0.50 | 0.30 | 0.25 |
| | 0.05 | 28.90 | 0.30 | 0.46 | 0.50 | 1.20 | 0.70 | 0.63 |
| | 0.10 | 9.14 | 0.98 | 1.24 | 0.92 | 1.76 | 0.58 | 1.10 |
| ILA+ TAIG-S | 0.03 | 63.34 | 0.32 | 0.72 | 0.86 | 2.34 | 1.10 | 1.07 |
| | 0.05 | 54.28 | 1.50 | 2.72 | 2.06 | 5.56 | 1.86 | 2.74 |
| | 0.10 | 25.16 | 3.70 | 6.00 | 2.36 | 6.92 | 1.68 | 4.13 |
| ILA+ TAIG-R | 0.03 | **69.40** | **0.60** | **1.08** | **1.10** | **3.38** | **1.56** | **1.54** |
| | 0.05 | **62.92** | **2.74** | **5.38** | **3.58** | **9.36** | **2.80** | **4.77** |
| | 0.10 | **34.44** | **6.76** | **14.08** | **4.24** | **11.20** | **2.68** | **7.79** |
| LinBP+ IFGSM | 0.03 | **98.68** | 0.04 | 0.10 | 0.28 | 1.02 | 0.58 | 0.40 |
| | 0.05 | **99.60** | 0.60 | 1.42 | 2.36 | 5.90 | 3.14 | 2.68 |
| | 0.10 | **99.60** | 4.84 | 7.96 | 9.54 | 17.52 | 9.18 | 9.81 |
| Linbp+ TAIG-S | 0.03 | 67.40 | 0.50 | 0.72 | 0.74 | 3.90 | 1.78 | 1.53 |
| | 0.05 | 68.66 | 3.82 | 4.60 | 5.40 | 13.56 | 6.20 | 6.72 |
| | 0.10 | 65.42 | **11.50** | **15.28** | **11.02** | **23.20** | **12.32** | **14.66** |
| Linbp+ TAIG-R | 0.03 | 62.54 | **1.18** | **1.56** | **1.80** | **6.64** | **3.48** | **2.93** |
| | 0.05 | 62.62 | **5.40** | **7.06** | **6.74** | **15.70** | **8.20** | **8.62** |
| | 0.10 | 51.98 | 10.34 | 13.58 | 9.30 | 19.90 | 10.76 | 12.78 |
| DI+ MIFGSM | 0.03 | **99.84** | 0.02 | 0.02 | 0.02 | 0.08 | 0.16 | 0.06 |
| | 0.05 | **100.00** | 0.02 | 0.10 | 0.20 | 0.28 | 0.18 | 0.16 |
| | 0.10 | **100.00** | 0.28 | 0.72 | 0.88 | 1.66 | 0.56 | 0.82 |
| DI+ MTAIG-S | 0.03 | 98.22 | 0.20 | 0.18 | 0.30 | 1.40 | 0.84 | 0.58 |
| | 0.05 | 98.36 | 0.84 | 1.02 | 1.36 | 4.50 | 1.94 | 1.93 |
| | 0.10 | 98.32 | 3.22 | 5.02 | 5.00 | 12.50 | 4.28 | 6.00 |
| DI+ MTAIG-R | 0.03 | 97.66 | **0.92** | **1.36** | **1.34** | **5.08** | **2.54** | **2.25** |
| | 0.05 | 98.30 | **4.94** | **7.22** | **6.46** | **18.42** | **8.40** | **9.09** |
| | 0.10 | 98.14 | **17.58** | **29.40** | **18.88** | **38.66** | **18.70** | **24.64** |

Table 15: The detection rate(%) and attack success rate after detection (ASRD %) of different methods.

| Detector | TWS | | FS | |
|---|---|---|---|---|
| Method | Detection rate | ASRD | Detection rate | ASRD |
| AOA | 5.72 | 31.66 | 4.28 | 4.71 |
| SI | 14.8 | 41.72 | 10.02 | 20.20 |
| NI-VT | 16.04 | 54.26 | 7.02 | 51.82 |
| LinBP | 49.46 | 31.36 | 15.04 | 57.58 |
| TAIG-S | 21.24 | 46.24 | 11.98 | 29.86 |
| TAIG-R | 21.16 | 68.52 | 12.22 | 67.58 |

number of sampling points, the attack success rate of TAIG-S only slightly fluctuates. Thus, we stop it at 70 sampling points and the results are listed in Table 16. For TAIG-R, we follow the same setting. The number of turning points $E$ also starts from 20 with an increment of 10 in each of the tests. And each of the segments in the path is estimated by one sampling point. We find that the attack success rate is slightly improved when the number of turning points increases. But the improvement becomes slow when the number of turning points is larger than 50. The results are listed in Table 17. To study the influence of the number of sampling points $S$ in each line segment, we fix the turning points $E$ to 20 and change $S$ from 1 to 5. With the increase of sampling points, the success attack success rate has slight improvements. The results are listed in Table 18.

## A.7 PERCEPTUAL EVALUATION

In this section, we provide the perceptual evaluation of different methods on seven full reference objective quality metrics, namely: Root Mean Square Error (RMSE), $L_0$ distance, Peak-Signal-to-Noise-Ratio (PSNR), Structural Similarity Index (SSIM) (Wang et al., 2004), Visual Information

Table 16: Attack success rate(%) of TAIG-S under different $S$ in the untargeted setting. The symbol $*$ indicates the surrogate model for generating adversarial examples.

| $S$ | ResNet$^*$ | Inception | PNASNet | SENet | DenseNet | MobileNet | Average |
|-----|-----------|-----------|---------|-------|----------|-----------|---------|
| 20 | 100.00 | 25.00 | 15.50 | 23.70 | 50.40 | 52.80 | 33.48 |
| 30 | 100.00 | 25.70 | 15.40 | 23.50 | 50.70 | 52.70 | 33.60 |
| 40 | 100.00 | 25.70 | 14.60 | 25.60 | 50.80 | 52.80 | 33.90 |
| 50 | 100.00 | 25.30 | 15.40 | 24.40 | 49.60 | 51.80 | 33.30 |
| 60 | 100.00 | 25.80 | 14.70 | 22.80 | 50.90 | 53.00 | 33.44 |
| 70 | 100.00 | 24.70 | 15.60 | 24.00 | 49.20 | 52.90 | 33.28 |

Table 17: Attack success rate(%) of TAIG-R under different $E$ in the untargeted setting. The symbol $*$ indicates the surrogate model for generating adversarial examples.

| $E$ | ResNet$^*$ | Inception | PNASNet | SENet | DenseNet | MobileNet | Average |
|-----|-----------|-----------|---------|-------|----------|-----------|---------|
| 20 | 100.00 | 44.50 | 34.70 | 42.10 | 71.20 | 72.60 | 53.02 |
| 30 | 100.00 | 45.50 | 36.20 | 44.60 | 73.10 | 73.70 | 54.62 |
| 40 | 100.00 | 46.60 | 36.10 | 43.90 | 74.20 | 75.00 | 55.16 |
| 50 | 100.00 | 47.50 | 36.40 | 45.70 | 75.70 | 75.30 | 56.12 |
| 60 | 100.00 | 48.30 | 36.80 | 44.80 | 75.30 | 75.90 | 56.22 |
| 70 | 100.00 | 47.80 | 37.30 | 44.50 | 75.60 | 75.00 | 56.04 |
| 80 | 100.00 | 47.50 | 37.40 | 46.10 | 76.10 | 75.50 | 56.52 |
| 90 | 100.00 | 48.30 | 37.50 | 45.70 | 76.70 | 76.10 | 56.86 |
| 100 | 100.00 | 47.40 | 37.50 | 45.90 | 77.20 | 76.30 | 56.86 |
| 150 | 100.00 | 47.70 | 37.90 | 46.00 | 76.30 | 76.40 | 56.86 |
| 200 | 100.00 | 47.40 | 37.90 | 46.00 | 77.40 | 75.70 | 56.88 |

Fidelity (VIFP) (Sheikh & Bovik, 2006), Multi-Scale SSIM index (MS-SSIM) (Wang et al., 2003), Universal Quality Index (UQI) (Wang & Bovik, 2002). 1000 images are used in this evaluation. The results are listed in Table 19. It shows that these methods perform similarly and TAIG-S is slightly better than the others.

Table 18: Attack success rate(%) of TAIG-R under different $S$ for $E = 20$ in the untargeted setting. The symbol $*$ indicates the surrogate model for generating adversarial examples.

| $S$ | ResNet$^*$ | Inception | PNASNet | SENet | DenseNet | MobileNet | Average |
|-----|-----------|-----------|---------|-------|----------|-----------|---------|
| 1 | 100.00 | 44.50 | 34.70 | 42.10 | 71.20 | 72.60 | 53.02 |
| 2 | 100.00 | 45.70 | 34.10 | 41.90 | 73.10 | 72.60 | 53.48 |
| 3 | 100.00 | 45.60 | 35.60 | 43.90 | 74.00 | 74.10 | 54.64 |
| 4 | 100.00 | 45.40 | 34.80 | 43.30 | 74.20 | 74.00 | 54.34 |
| 5 | 100.00 | 46.90 | 35.20 | 44.80 | 75.50 | 74.40 | 55.36 |

Table 19: Perceptual evaluation of different methods on reference objective quality metrics. The upward pointing arrow indicates that higher is better and the downward pointing arrow indicates that lower is better.

| Methods | $\varepsilon$ | $L_0\downarrow$ | SSIM↑ | RMSE↓ | PSNR↑ | UQI↑ | VIFP↑ | MS-SSIM↑ |
|---------|------|------|------|------|------|------|------|------|
| AOA | 8/255 | 0.968 | 0.913 | 0.027 | 31.36 | 0.973 | 0.988 | 0.969 |
| | 16/255 | 0.979 | 0.751 | 0.054 | 25.41 | 0.948 | 0.973 | 0.910 |
| SI | 8/255 | 0.973 | 0.921 | 0.026 | 31.65 | 0.975 | 0.981 | 0.971 |
| | 16/255 | 0.977 | 0.779 | 0.050 | 25.98 | 0.952 | 0.968 | 0.918 |
| NI-VT | 8/255 | 0.958 | 0.937 | 0.024 | 32.49 | 0.975 | 0.993 | 0.969 |
| | 16/255 | 0.966 | 0.824 | 0.045 | 27.02 | 0.955 | 0.987 | 0.911 |
| LinBP | 8/255 | 0.978 | 0.919 | 0.026 | 31.55 | 0.975 | 0.967 | 0.972 |
| | 16/255 | 0.983 | 0.767 | 0.052 | 25.76 | 0.950 | 0.923 | 0.914 |
| TAIG-S | 8/255 | **0.921** | **0.940** | **0.023** | **32.66** | **0.981** | **0.991** | **0.978** |
| | 16/255 | **0.937** | **0.846** | **0.041** | **27.67** | **0.965** | **0.995** | **0.942** |
| TAIG-R | 8/255 | 0.952 | 0.931 | 0.025 | 31.96 | 0.976 | 0.987 | 0.968 |
| | 16/255 | 0.968 | 0.805 | 0.049 | 26.16 | 0.953 | 0.997 | 0.906 |

## A.8 VISUALIZATION OF IG AND RIG

In this section, we provide more IG and RIG of different images. Fig. 5 is original images used in the visualization. Fig. 6 and Fig. 7 are the IG and RIG of Fig. 5(a) and Fig. 5(b) from different networks. Fig. 8 and Fig. 9 are the IG and RIG of Fig. 5(c) and Fig. 5(d) before and after attacks.

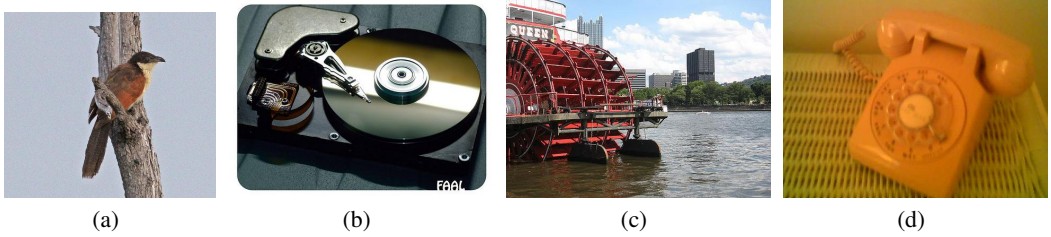

(a)  (b)  (c)  (d)

Figure 5: Original images.

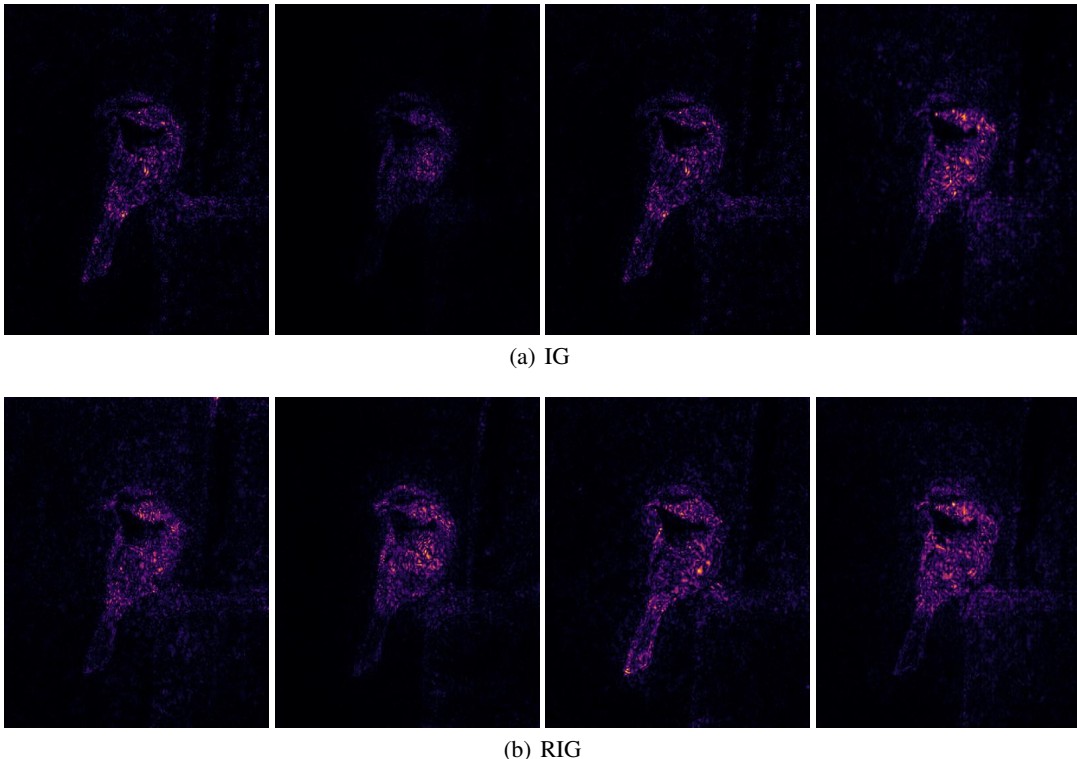

(a) IG

(b) RIG

Figure 6: The IG and RIG of Fig. 5(a) from ResNet, MobileNet, SENet and Inception, from left to right.

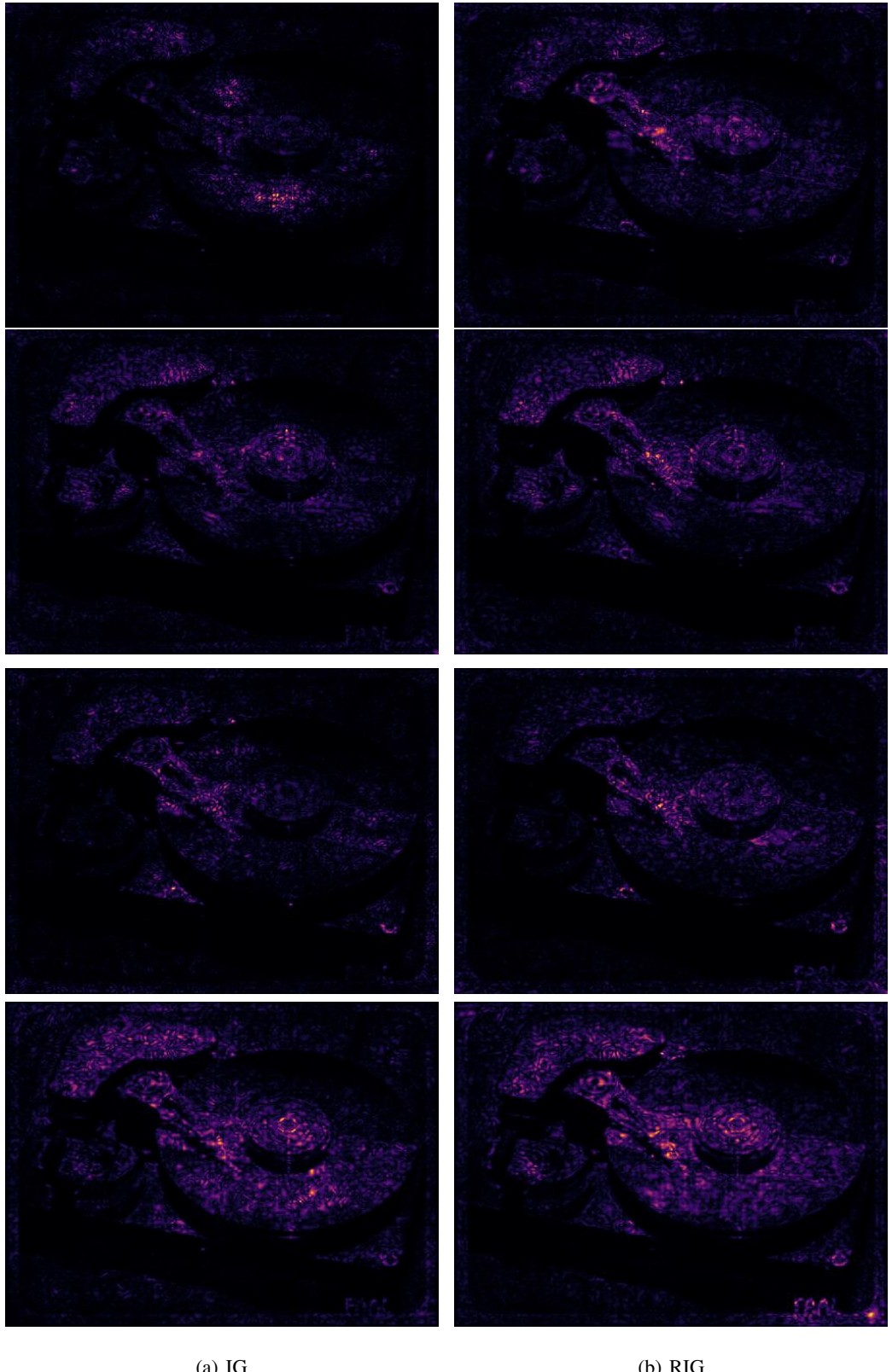

(a) IG                                   (b) RIG

Figure 7: The IG and RIG of Fig. 5(b) from ResNet, MobileNet, SENet and Inception, from top to bottom.

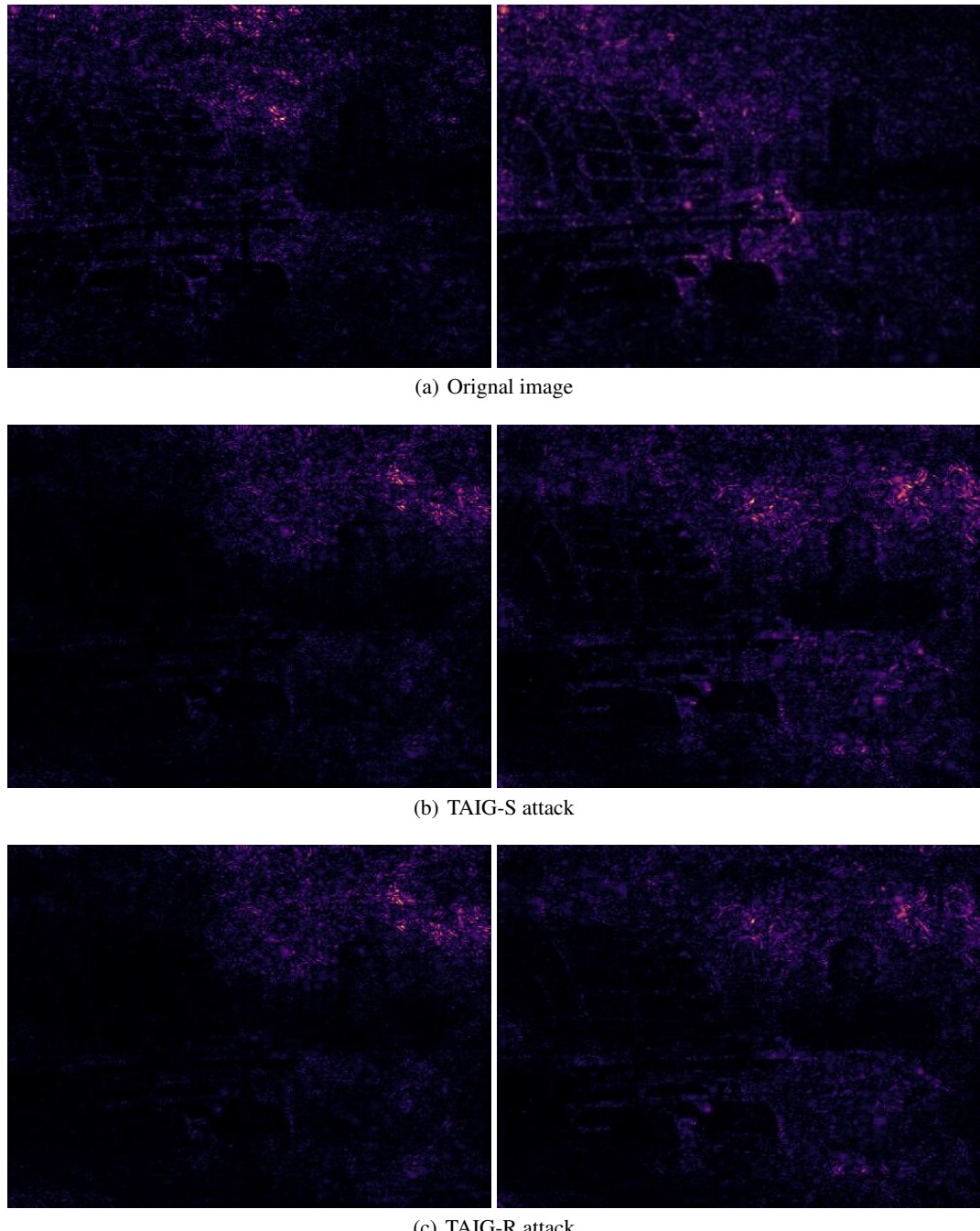

(a) Orignal image

(b) TAIG-S attack

(c) TAIG-R attack

Figure 8: The IG and RIG of (a) the original image in Fig. 5(c), (b) the image after TAIG-S attack and (c) the image after TAIG-R attack. The left column is IG and the right column is RIG.

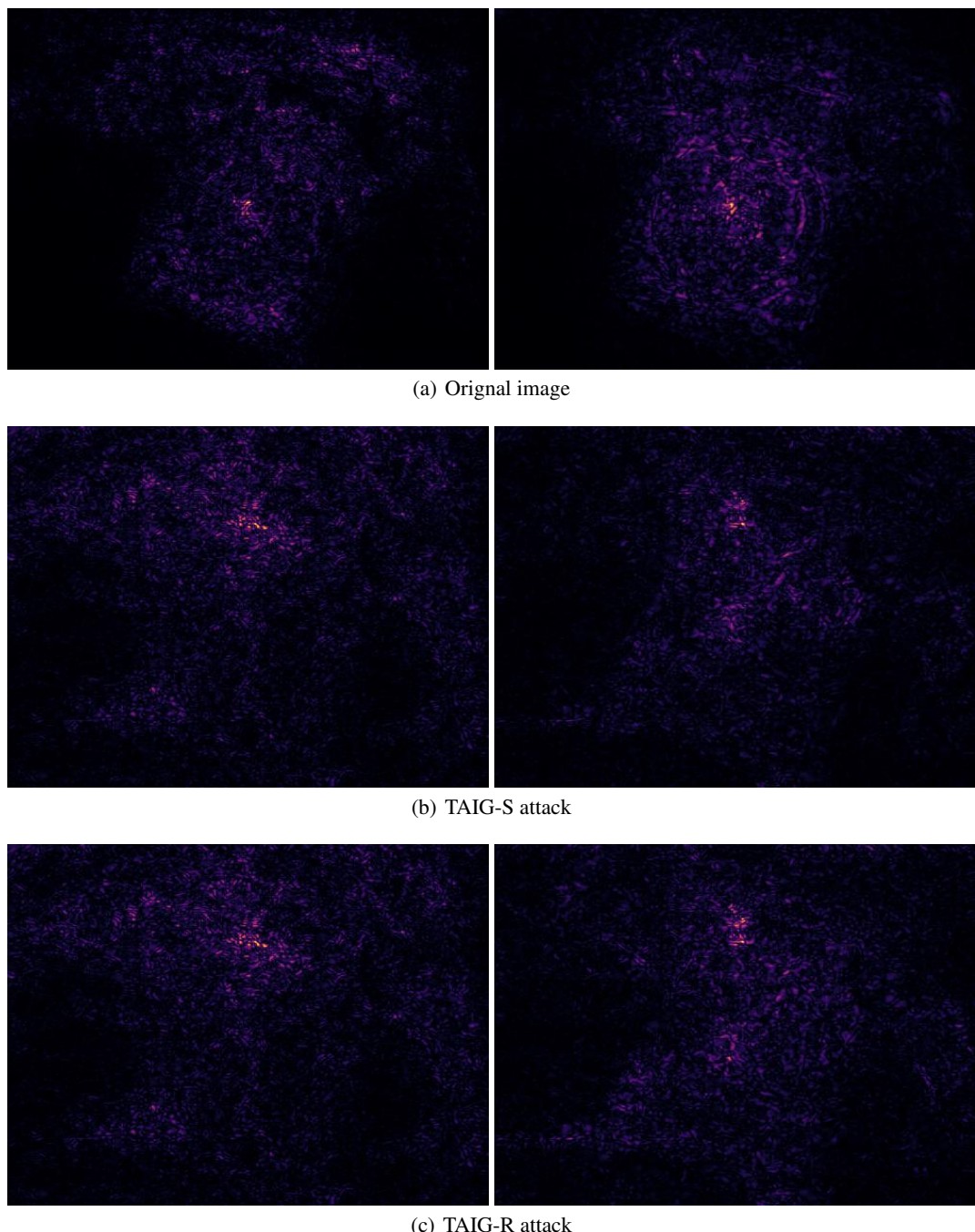

(a) Orignal image

(b) TAIG-S attack

(c) TAIG-R attack

Figure 9: The IG and RIG of (a) the original image in Fig. 5(d), (b) the image after TAIG-S attack and (c) the image after TAIG-R attack. The left column is IG and the right column is RIG.

