# OpenReview forum: "Transferable Adversarial Attack based on Integrated Gradients"
_ICLR.cc/2022/Conference — ICLR 2022 Poster_

### Official Review · Reviewer_xf97 · 2021-10-30

**Correctness:** 3
**Technical Novelty And Significance:** 2
**Empirical Novelty And Significance:** 3
**Recommendation:** 5
**Confidence:** 3

**Main Review:**

Strengths:
1. The experimental results are good.
2. The idea of updating along the integrated gradients is new and has theoretical support.

Weaknesses:
1. The most important weakness is the confusing experimental settings. Firstly, this paper lacks the details of calculating integrated gradients, i.e., how to calculate Equation (1)? According to the experiments section, I guess the integrated gradients are computed by averaging the gradients of 30 sampling points. Thus, the TAIG-S is 30X slower than most methods with the same iterations, such as LinBP and AOA. Although SI also requires calculating multiple gradients, TAIG-S is still 6X slower than SI (with the number of scale copies of 5). Furthermore, TAIG-R requires more gradient calculations because it should calculate function IG(f,x) E (the number of turning points) times. However, the authors did not report the value of E in the experiment section. Thus, I think it is better to compare the performance with other methods under the same gradient calculations instead of iterations.
2. In section 3.3, the authors claim that sign(IG(f,x)) approximates sign(\nabla f). I think it is a very weak approximation since there are only 68% elements of sign(IG(f,x)) and sign(\nabla f) are the same, while random guessing can achieve about 50% of the same elements as sign(\nabla f). The approximation of RIG is much worse (only 58%). Is this because the limited number of sampling points? Will the similarity between sign(IG(f,x)) and sign(\nabla f) be higher if more sampling points are used to calculate IG? Besides, TAIG-S approximates sign(\nabla f) better than TAIG-R while TAIG-R is better than TAIG-S in all experimental results. How do you explain this phenomenon?
3. Ablation studies are missing. It’s necessary to see the sensitivity of the number of sampling points (30 in this paper) and the number of random paths (E, which is not be reported in this paper).
4. From Table 1, LinBP seems to be the strongest counterpart. Why not compare with LinBP when VGG19 (or the other surrogate model listed in Table 4) is taken as the surrogate model? Finally, as I said before, LinBP only calculates gradients once in each iteration, it is unfair to compare it with TAIG-R which calculates gradients 30E times in each iteration. I recommend the authors to compare with some advanced methods which require more gradient calculations [1,2, 3].
5. Maybe a typo: What’s the name of your methods? In the abstract, the method is called Transferable Attack “based on” Integrated Gradients. In later, the name is changed to Transferable Attack “using” Integrated Gradients.

[1] Wu W, Su Y, Lyu M R, et al. Improving the Transferability of Adversarial Samples With Adversarial Transformations[C]//Proceedings of the IEEE/CVF Conference on Computer Vision and Pattern Recognition. 2021: 9024-9033.
[2] Wang X, He K. Enhancing the Transferability of Adversarial Attacks through Variance Tuning[C]//Proceedings of the IEEE/CVF Conference on Computer Vision and Pattern Recognition. 2021: 1924-1933.
[3] Wang X, He X, Wang J, et al. Admix: Enhancing the transferability of adversarial attacks[C]// Proceedings of the IEEE/CVF International Conference on Computer Vision (ICCV). 2021: 16158-16167.


**Summary Of The Paper:**

Two methods are proposed in this paper. They are Transferable Attack using Integrated Gradients on Straight-line Path (TAIG-S) and Transferable Attack using Integrated Gradients on Random Piecewise Linear Path (TAIG-R). Compared with typical gradient-based attack methods, the TAIG-S uses integrated gradients to update adversarial examples, and TAIG-R uses random path integrated gradients to update adversarial examples. Experiments on ImageNet can prove the effectiveness of these methods.

**Summary Of The Review:**

This paper proposes a novel algorithm to generate adversarial examples, and the results are good. However, some analysis and experimental results are confusing. The most important weakness is the high computational cost. Please address issues listed in the detailed comments during the rebuttal and make the paper clearer.

---

> ### Author Response · Authors · 2021-11-19
> **response to reviewer xf97 on Q1 and Q5**
>
> We would like to thank reviewer xf97 for the thoughtful comments. Below we address the concerns mentioned in the review:
>
> Q1.1: The paper lacks the details of calculating integrated gradients and the value of E.
>
> We described the discrete version of IG in the last paragraph in page 5 ( “Therefore, $\mathrm{sign}(\mathrm{IG}(f_y,\boldsymbol{x}))$ in equation 3 only depends on $\int_{\eta=0}^{1}\frac{\partial f(\eta\boldsymbol{x})}{\partial x_i}\mathrm{d}\eta$, whose discrete version is $\frac{1}{S}\sum_{s=0}^{S}\frac{\partial f(s\times\boldsymbol{x}/S)}{\partial x_i}$.”). For the value of $E$, we gave it in the last few sentences on page 6 (“Thirty sampling points are used to estimate TAIG-S and TAIG-R. For TAIG-R, each segment is estimated by one sampling point ”). In other words, E is thirty. We realized that the statement is not clear enough, so we changed it to "Thirty sampling points are used to estimate TAIG-S. For TAIG-R, the number of turning points $E$ is set to 30, and $\tau$ is set equal to $\varepsilon$. Because the line segment is short, each segment is estimated by one sampling point in TAIG-R."
>
> Q1.2: I think it is better to compare the performance with other methods under the same gradient calculations instead of iterations.
>
> Thanks for your comments. We have run LinBP and SI with the same number of gradient calculations as the proposed method under $\varepsilon=4/255,8/255/,16/255$. For AOA, the calculation of SGLPR is too slow, and we found that the average attack success rate of AOA only changes from 32.07 (100 iterations) to 32.19 (300 iterations) under $\varepsilon=16/255$. Because we have limited computing resources and the result of AOA is not as good as LinBP and SI, we did not run it for more iterations. We have added these experimental results in the supplemental material based on your comments . We listed the results of LinBP and SI with the same number of gradient calculations in the table below:
>
> | Method | $\varepsilon$ | Gradient calculation number | ResNet | Inception | PNASNet | SENet | DenseNet | MobileNet | Average |
> |:------:|:-----------:|:---------------------------:|:------:|:---------:|:-------:|:-----:|:--------:|:---------:|:-------:|
> | Linbp  |    4/255    |             600             |  99.98 |    8.70   |   6.30  | 11.70 |   24.72  |   26.62   |  15.61  |
> |        |    8/255    |             1500            |    100.00 |   29.52   |  27.10  | 43.16 |   64.12  |   63.22   |  45.42  |
> |        |    16/255   |             3000            |   100.00 |   74.02   |  74.50  | 87.48 |   94.86  |   94.68   |  85.11  |
> |   SI   |    4/255    |             600             |     99.90 |    8.00   |   3.50  |  5.12 |   16.94  |   18.78   |  10.47  |
> |        |    8/255    |             1500            |     100.00 |   20.62   |  10.82  | 16.08 |   39.60  |   42.26   |  25.88  |
> |        |    16/255   |             3000            |   100.00 |   48.34   |  35.74  | 48.60 |   73.78  |   73.18   |  55.93  |
> | TAIG-R |    4/255    |             600             |     99.96 |   17.04   |   8.66  | 11.60 |   35.38  |   37.28   |  21.99  |
> |        |    8/255    |             1500            |      100.00 |   45.44   |  34.00  | 41.62 |   73.72  |   73.86   |  53.73  |
> |        |    16/255   |             3000            |    100.00 |   84.30   |  78.82  | 84.72 |   97.00  |   95.90   |  88.15  |
>
> Q5: Maybe a typo: What’s the name of your methods? In the abstract, the method is called Transferable Attack “based on” Integrated Gradients. In later, the name is changed to Transferable Attack “using” Integrated Gradients.
>
> We apologize for the typo. We have corrected the method name to Transferable Attack based on Integrated Gradients in the paper. Thanks for you to point it out.

---

> ### Author Response · Authors · 2021-11-19
> **response to reviewer xf97 on Q2**
>
> Q2: In section 3.3, the authors claim that sign(IG(f,x)) approximates sign(\nabla f). I think it is a very weak approximation since there are only 68% elements of sign(IG(f,x)) and sign(\nabla f) are the same, while random guessing can achieve about 50% of the same elements as sign(\nabla f). The approximation of RIG is much worse (only 58%). Is this because the limited number of sampling points? Will the similarity between sign(IG(f,x)) and sign(\nabla f) be higher if more sampling points are used to calculate IG?  Besides, TAIG-S approximates sign(\nabla f) better than TAIG-R, How do you explain this phenomenon?
>
> We believe that the reviewer focuses on the number, 68%, and the word, approximation. First of all, we would like to highlight that the mathematical discussion is not to prove that $\mathrm{sign}({\nabla f})$ and $\mathrm{sign}(\boldsymbol{\mathrm{IG}}(\boldsymbol{x}))$  are highly similar. It is to indicate that they have some similarity and where their difference comes from. As mentioned in the original paper, “the quality of this approximation depends on Eq. 12”, which is
> \begin{equation}
> 	\frac{\partial\mathrm{IG}_i(\boldsymbol{x})}{\partial x_i}\approx\frac{\mathrm{IG}_i(\boldsymbol{x})-\mathrm{IG}_i(\boldsymbol{x}-h\Delta\boldsymbol{x})}{h}
> \end{equation}
> Let us draw an analogy between the first-order Taylor approximation and our mathematical discussion. It would be easier to explain it. The first-order Taylor approximation is given below
> \begin{equation}
>   g(x) \approx g(a)+g^{'}(a)(x-a).
> \end{equation}
> It is well-known that the quality of this approximation depends on the difference between $x$ and $a$. When $a$ is far away from $x$, the quality of the approximation would drop. As with the first-order Taylor approximation, the quality of our approximation (Eq. 12) depends on $h$, which indicates the difference between $x$ and $x-h \Delta x$. In the derivation, we do not define $h$ explicitly because $\mathrm{IG}_i (x-h \Delta x)$ is set to zero. More clearly, the issue about the quality of approximation in Eq. 12 is the same as the first-order Taylor approximation. When two points are closer, the quality of the approximation will be better. However, to perform a transferable attack, $\mathrm{IG}_i (x-h \Delta x)$ is set to zero, which influences the quality of the approximation. The number of sampling points is not the major factor. It should be highlighted that a better approximation is not our goal. When $\mathrm{sign}(\boldsymbol{\mathrm{IG}}(\boldsymbol{x}))$  is very close to $\mathrm{sign}({\nabla f})$, it only implies that it is stronger for white-box attack. However, it is well-known that $\mathrm{sign}({\nabla f})$ is not ideal for the black-box attack, which is the target of this study. More clearly, we do not expect that $\mathrm{sign}(\boldsymbol{\mathrm{IG}}(\boldsymbol{x}))$   is very close to $\mathrm{sign}({\nabla f})$. To avoid confusion, in the revised version, remarks have been added and the term weakly approximate is used whenever is suitable. TAIG-S approximates $\mathrm{sign}({\nabla f})$  better than TAIG-R because TAIG-R adds noise in the integration path. This noise further degrades the quality of the approximation.

---

> ### Author Response · Authors · 2021-11-19
> **response to reviewer xf97 on Q4.1**
>
> Q4.1: why not compare LinBP when with VGG19 (or the other surrogate model listed in Table 4)
>
> LinBP is sensitive to the choice of the position from where the network is modified. The authors of LinBP reported their results on ImageNet and the corresponding modified layers in ResNet50. That's the reason why we compared with LinPB using ResNet50 as the surrogate model. Following your advice, we compared LinBP with TAIG-R under $\varepsilon=16/255$ on VGG19. 5000 images were used in this experiment. To identify the best position (layer) to start the model modification, we tested each position in VGG19 (16 in total) separately and generated adversarial examples from these 16 different models with 100 iterations. Then we used the six black-box models mentioned in Table 1 to compare their performances. The results are listed in the table below:
>
> | Position | ResNet | Inception | PNASNet | SENet | DenseNet | MobileNet | Average |
> |:-----------:|:------:|:---------:|:-------:|:-----:|:--------:|:---------:|:-------:|
> |     1    |  35.71 |   20.67   |   9.19  | 17.76 |   24.52  |   47.48   |  25.89  |
> |     2    |  37.81 |   19.76   |  10.19  | 19.05 |   25.92  |   49.38   |  27.02  |
> |     3    |  44.67 |   20.38   |  10.10  | 20.86 |   27.05  |   54.57   |  29.61  |
> |     4    |  45.29 |   14.14   |   7.00  | 18.86 |   23.05  |   51.81   |  26.69  |
> |     5    |  40.57 |   18.33   |  10.57  | 22.00 |   27.95  |   55.24   |  29.11  |
> |     6    |  46.71 |   24.52   |  11.19  | 22.38 |   38.71  |   65.62   |  34.86  |
> |     7    |  25.57 |   22.24   |  13.29  | 15.71 |   28.19  |   54.21   |  26.54  |
> |     8    |  32.14 |   19.05   |  11.33  | 21.29 |   28.05  |   53.81   |  27.61  |
> |     9   |  23.71 |   16.86   |  20.05  | 31.57 |   54.76  |   27.95   |  29.15  |
> |    10    |  74.90 |   56.10   |  53.48  | 58.90 |   70.48  |   82.29   |  66.03  |
> |    11    |  30.62 |   23.10   |  16.48  | 24.14 |   32.19  |   51.43   |  29.66  |
> |    12    |  66.71 |   47.19   |  45.57  | 56.00 |   63.76  |   75.24   |  59.08  |
> |    13    |  51.71 |   35.24   |  32.00  | 46.10 |   51.57  |   65.52   |  47.02  |
> |    14    |  74.43 |   49.10   |  58.57  | 67.10 |   71.38  |   81.10   |  66.95  |
> |    15    |  62.19 |   37.81   |  43.33  | 53.76 |   59.00  |   74.52   |  55.10  |
> |    16    |  69.52 |   43.57   |  53.00  | 59.52 |   66.10  |   77.52   |  61.54  |
>
> To perform further evaluation, we selected two modified models with the best performance and used them to generate adversarial examples with 3000 iterations. The results are listed in the table below. We have added these experimental results in section A.3.2 in the appendix.
>
>
> |   LinBP position | ResNet | Inception | PNASNet | SENet | DenseNet | MobileNet | Average |
> |:---------:|:------:|:---------:|:-------:|:-----:|:--------:|:---------:|:-------:|
> |   TAIG-R  |  93.96 |   84.70   |  86.20  | 86.88 |   93.64  |   96.24   |  90.27  |
> | LinBP(10) |  83.16 |    67.7   |  64.76  | 67.68 |   80.74  |   88.12   |  75.36  |
> | LinBP(14) |  79.32 |   51.58   |   59.8  | 73.42 |   74.32  |    85.7   |  70.69  |

---

> ### Author Response · Authors · 2021-11-19
> **response to reviewer xf97 on Q4.2**
>
> Q 4.2: I recommend the authors compare with some advanced methods which require more gradient calculations [1,2, 3].
>
> Thanks for your advice. We checked the references [1,2,3]. As [1] does not provide the source code, we can only compare with [2] and [3]. [2] and [3] both sampled 1000 images from ImageNet and used $\varepsilon=16/255$. Their surrogate models are InceptionV3. Thus, we ran TAIG-R on InceptionV3 on the same 1000 images they used under the same value of $\varepsilon$. All the methods ran with the same number of gradient calculations (3000). Below are the experimental results.
>
> | Methods | Inception(*) | PNASNet | SENet | DenseNet | MobilenNet | ResNet | Average |
> |------------|:------------:|:-------:|:-----:|:--------:|:----------:|:------:|:-------:|
> | MI_Admix [3]   | 99.1         | 51.8    | 55.1  | 67.8     | 69.6       | 67.7   | 62.4    |
> | VMI [2]        | 98.8         | 60.4    | 60.4  | 68.5     | 70.9       | 69.6   | 65.96   |
> | VNI [2]        | 98.9         | 61.4    | 62.1  | 68.6     | 71.8       | 70.7   | 66.92   |
> | TAIG-R     | 100          | 65.3    | 64.6  | 79.9     | 82.9       | 79.8   | 74.5    |
>
> From the table, we can see that TAIG-R has a higher success attack rate than the methods proposed by [2] and [3] and VNI performs better than VMI and MI_Admix. Thus, we ran VNI on our dataset using ResNet50 as the surrogate model under different $\varepsilon$ and the results are listed below. We have added this comparison in section A.3.3 in the appendix.
>
> | Method | $\varepsilon$ | Gradient calculation number   | ResNet | Inception | Pnasnet | Senet | Densenet | Mobilenet | Average |
> |:------:|:-------------:|:-----------------------------:|:------:|:---------:|:-------:|:-----:|:--------:|:---------:|:-------:|
> |   VNI  |      0.03     |              600              |  95.40 |   32.94   |  18.62  | 18.90 |   38.40  |   42.68   |  30.31  |
> |        |      0.05     |              1500             |  95.80 |   52.70   |  36.16  | 34.86 |   56.94  |   61.14   |  48.36  |
> |        |      0.1      |              3000             |  96.70 |   79.88   |  71.42  | 68.23 |   83.20  |   84.66   |  77.48  |
> | TAIG-R   |  0.03  |  600 | 100.00 | 46.06 | 33.82 | 40.14 | 72.52 | 71.58 | 52.82 |
> |         |  0.05  | 1500 | 100.00 | 74.22 | 66.30 | 73.46 | 93.62 | 91.56 | 79.83 |
> |         |   0.1  | 3000 | 100.00 | 95.18 | 94.50 | 96.02 | 99.60 | 99.20 | 96.90 |
>
>
> [1] Wu W, Su Y, Lyu M R, et al. Improving the Transferability of Adversarial Samples With AdversarialTransformations[C]//Proceedings of the IEEE/CVF Conference on Computer Vision and Pattern Recognition.2021: 9024-9033.
>
> [2] Wang X, He K. Enhancing the Transferability of Adversarial Attacks through VarianceTuning[C]//Proceedings of the IEEE/CVF Conference on Computer Vision and Pattern Recognition. 2021: 1924-1933.
>
> [3] Wang X, He X, Wang J, et al. Admix: Enhancing the transferability of adversarial attacks[C]//Proceedings of the IEEE/CVF International Conference on Computer Vision (ICCV). 2021: 16158-16167.

---

> ### Author Response · Authors · 2021-11-19
> **response to reviewer xf97 on Q3**
>
> Q3: Ablation studies are missing. It’s necessary to see the sensitivity of the number of sampling points and the number of random paths.
>
> About the number of sampling points, we followed the guideline given in [4]. Sundararajan et al. pointed out that the sampling points between 20 to 300 are enough to approximate the integral. To balance the computation cost and attack success rate, we used 30 sampling points in the experiments for TAIG-S. For TAIG-R, $E$ is set to 30 and because the line segment is short, each segment is estimated by one sampling point.
>
> Following your advice, we added an ablation study about the number of sampling points in TAIG-S and TAIG-R and added this result in section A.6 in the appendix.  $\varepsilon=8/255$ and 1000 images are used in the ablation study. According to the suggestion in [4], the number of sampling points for TAIG-S starts from 20 with an increment of 10 in each of the tests. We found that with the increase of the number of sampling points, the attack success rate only slightly fluctuates. So, we stopped at 70 sampling points and the results are listed below:
>
> | sampling points | ResNet | Inception | PNASNet | SENet | DenseNet | MobileNet | Average |
> |:-----:|--------|-----------|---------|-------|----------|-----------|---------|
> | 20    | 100    | 25        | 15.5    | 23.7  | 50.4     | 52.8      | 33.48   |
> | 30    | 100    | 25.7      | 15.4    | 23.5  | 50.7     | 52.7      | 33.60   |
> | 40    | 100    | 25.7      | 14.6    | 25.6  | 50.8     | 52.8      | 33.90   |
> | 50    | 100    | 25.3      | 15.4    | 24.4  | 49.6     | 51.8      | 33.30   |
> | 60    | 100    | 25.8      | 14.7    | 22.8  | 50.9     | 53        | 33.44   |
> | 70    | 100    | 24.7      | 15.6    | 24    | 49.2     | 52.9      | 33.28   |
>
>
> For TAIG-R, we used the same setting. The number of turning points ($E$) is set to start from 20 with an increment of 10 in each of the tests. And each of the segments in the path is estimated by one sampling point. We found that the increase of the number of turning points slightly improves the attack success rate. But the improvement becomes slow when $E$ is larger than 50. The results are listed below:
>
> | $E$ | ResNet | Inception | PNASNet | SENet | DenseNet | MobileNet | Average |
> |:-----:|:------:|:---------:|:-------:|:-----:|:--------:|:---------:|:-------:|
> |   20  | 100    | 44.5      | 34.7    | 42.1  | 71.2     | 72.6      |  53.02  |
> |   30  | 100    | 45.5      | 36.2    | 44.6  | 73.1     | 73.7      |  54.62  |
> |   40  | 100    | 46.6      | 36.1    | 43.9  | 74.2     | 75        |  55.16  |
> |   50  | 100    | 47.5      | 36.4    | 45.7  | 75.7     | 75.3      |  56.12  |
> |   60  | 100    | 48.3      | 36.8    | 44.8  | 75.3     | 75.9      |  56.22  |
> |   70  | 100    | 47.8      | 37.3    | 44.5  | 75.6     | 75        |  56.04  |
> |   80  | 100    | 47.5      | 37.4    | 46.1  | 76.1     | 75.5      |  56.52  |
> |   90  | 100    | 48.3      | 37.5    | 45.7  | 76.7     | 76.1      |  56.86  |
> |  100  | 100    | 47.4      | 37.5    | 45.9  | 77.2     | 76.3      |  56.86  |
> |  150  | 100  |   47.70   |  37.90  | 46.00 |   76.30  |   76.40   |  56.86  |
> |  200  | 100 |   47.40   |  37.90  | 46.00 |   77.40  |   75.70   |  56.88  |
>
>
> To estimate the influence of the number of sampling points in each line segment, we fixed the turning points $E$ to 20 and changed the sampling points from 1 to 5. With the increase of the number of sampling points, the attack success rate gets slightly improvement. The results are listed below:
>
> | Sampling points| ResNet | Inception | PNASNet | SENet | DenseNet | MobileNet | Average |
> |------------|--------|-----------|---------|-------|----------|-----------|---------|
> | 1          | 100    | 44.5      | 34.7    | 42.1  | 71.2     | 72.6      | 53.02   |
> | 2          | 100    | 45.7      | 34.1    | 41.9  | 73.1     | 72.6      | 53.48   |
> | 3          | 100    | 45.6      | 35.6    | 43.9  | 74       | 74.1      | 54.64   |
> | 4          | 100    | 45.4      | 34.8    | 43.3  | 74.2     | 74        | 54.34   |
> | 5          | 100    | 46.9      | 35.2    | 44.8  | 75.5     | 74.4      | 55.36   |
>
> [4] Mukund, Sundararajan, Ankur Taly, and Qiqi Yan. "Axiomatic attribution for deep networks." International Conference on Machine Learning, 2017.

---

### Official Review · Reviewer_N6RH · 2021-11-02

**Correctness:** 3
**Technical Novelty And Significance:** 3
**Empirical Novelty And Significance:** 3
**Recommendation:** 6
**Confidence:** 3

**Main Review:**

The authors propose a simple but effective technique for generating adversarial perturbations in a black-box setting. In addition to this, the core component of the proposed technique (the Integrated Gradients and the Random Path Integrated Gradients) can be used to improve the existing techniques. I found the paper to be clear. I would encourage the authors to additional experiments to justify the claims made in the paper.

Below are a few questions for the authors:
How does your method stack up against other classes of black-box adversarial attacks? For instance, I would encourage the authors to compare against gradient-free attacks.
How imperceptible are the examples generated by the proposed technique? For instance, even though most black-box adversarial have higher success rates, they generally forgo imperceptibility.
How good is the proposed technique in evading the recent class of adversarial example detection methods?

I would also encourage the authors to include a detailed set of ablation studies, such as how many samples are needed to ensure that the surrogate models are sufficient. Does TAIG require less in comparison to others? Also, over how many samples are the Integrated Gradients computed, does that impact your results? In addition to the above, it will also be helpful to report run-times for each of the techniques.

**Summary Of The Paper:**

In practice, adversarial examples are generated in three ways, i) solving a standard optimisation problem, ii) leveraging the salient regions of an image, or iii) smoothing the decision surfaces. The authors propose a simple technique named Transferable Attack based on Integrated Gradients (TAIG) that combines all these three approaches. Unlike the existing systems, which leverages other methods by adding additional terms to the objective function, TAIG integrates them into a single objective.

**Summary Of The Review:**

In general, the paper explores an exciting direction, but I found the experiments a bit lacking. I would encourage the authors to address the questions listed above

---

> ### Author Response · Authors · 2021-11-19
> **response to reviewer N6RH on Q1**
>
> We would like to thank reviewer N6RH for your comments and suggestions. Below we address the concerns/questions mentioned in the review:
>
> Q1: How does your method stack up against other classes of black-box adversarial attacks? For instance, I would encourage the authors to compare against gradient-free attacks.
>
> Transfer-based attacks, which the proposed method belongs to, and gradient-free attacks [1,2,3] are under different assumptions. Usually,  gradient-free attacks attack a black-box model by a huge number of queries while transfer-based attacks pass adversarial examples to the model once. The previous transfer attack studies, including [4-11],  did not compare methods across these two categories, because they are studied under different threat models.
>
> [1] Brunner, Thomas, et al. "Guessing smart: Biased sampling for efficient black-box adversarial attacks." Proceedings of the IEEE/CVF International Conference on Computer Vision. 2019.
>
> [2] Alzantot, Moustafa, et al. "Genattack: Practical black-box attacks with gradient-free optimization." Proceedings of the Genetic and Evolutionary Computation Conference. 2019.
>
> [3] Zhao, Pu, et al. "On the design of black-box adversarial examples by leveraging gradient-free optimization and operator splitting method." Proceedings of the IEEE/CVF International Conference on Computer Vision. 2019.
>
> [4] Yiwen Guo, Qizhang Li, and Hao Chen. Backpropagating linearly improves transferability of adversarial examples. In Advances in Neural Information Processing Systems, 2020.
>
> [5] Wu, Dongxian, et al. "Skip Connections Matter: On the Transferability of Adversarial Examples Generated with ResNets." International Conference on Learning Representations. 2019.
>
> [6] Jiadong Lin, Chuanbiao Song, Kun He, Liwei Wang, and John E. Hopcroft. Nesterov accelerated
> gradient and scale invariance for adversarial attacks. In International Conference on Learning
> Representations, 2020.
>
> [7] Wu, Weibin, et al. "Boosting the transferability of adversarial samples via attention." Proceedings of the IEEE/CVF Conference on Computer Vision and Pattern Recognition. 2020.
>
> [8] Inkawhich, Nathan, et al. "Feature space perturbations yield more transferable adversarial examples." Proceedings of the IEEE/CVF Conference on Computer Vision and Pattern Recognition. 2019.
>
> [9] Sizhe Chen, Zhengbao He, Chengjin Sun, Jie Yang, and Xiaolin Huang. Universal adversarial
> attack on attention and the resulting dataset damagenet. IEEE Transactions on Pattern Analysis
> and Machine Intelligence, 2020.
>
> [10] Wen Zhou, Xin Hou, Yongjun Chen, Mengyun Tang, Xiangqi Huang, Xiang Gan, and Yong Yang.
> Transferable adversarial perturbations. In Proceedings of the European Conference on Computer
> Vision, 2018.
>
> [11] Qian Huang, Isay Katsman, Horace He, Zeqi Gu, Serge Belongie, and Ser-Nam Lim. Enhancing
> adversarial example transferability with an intermediate level attack. In Proceedings of the IEEE
> International Conference on Computer Vision, 2019.

---

> ### Author Response · Authors · 2021-11-19
> **response to reviewer N6RH on Q2**
>
> Q2: How imperceptible are the examples generated by the proposed technique? For instance, even though most black-box adversarial have higher success rates, they generally forgo imperceptibility. How good is the proposed technique in evading the recent class of adversarial example detection methods?
>
> We have checked the recent adversarial example detection methods, including [12-25]. Among these detection methods, [12-20] were designed or evaluated on small image datasets such as CIFAR10 and MNIST. [17,21-25] provided evaluation on ImageNet. However, only [21] and [24] provided the source codes. Using their default settings, we compared the results of different attack methods under $\varepsilon= 16/255$. The detection rate and the attack success rate after detection (ASRD) are listed below. In this experiment, 5000 images were used and the ASRD is defined as the number of successful attacks over 5000.
>
> |     Detector         |        [16]        |                   |   [13]             |                   |
> |:----------:|----------------|-------------------|----------------|-------------------|
> | Method     | Detection rate | ASRD | Detection rate | ASRD |
> | AOA        | 5.72           | 31.66             | 4.28           | 4.71              |
> | VNI        | 16.04          | 54.26             | 7.02           | 51.82             |
> | LinBP      | 49.46          | 31.36             | 15.04          | 57.58             |
> | TAIG-S     | 21.24          | 46.24             | 11.98          | 29.86             |
> | TAIG-R     | 21.16          | 68.52             | 12.22          | 67.58             |
> | SI         | 14.8           | 41.72             | 10.02          | 20.20             |
>
> The detection rates of both detectors on AOA are low, but the SARD of AOA is the lowest one. From the table, we can see that TAIG-R has the highest ASRD among all the methods.
>
> [12] Grosse, K., Manoharan, P., Papernot, N., Backes, M., and McDaniel, P. On the statistical detection of adversarial examples. arXiv preprint arXiv:1702.06280, 2017.
>
> [13] Ma, Xingjun, et al. "Characterizing Adversarial Subspaces Using Local Intrinsic Dimensionality." International Conference on Learning Representations. 2018.
>
> [14] Yin, X., Kolouri, S., and Rohde, G. K. Gat: Generative adversarial training for adversarial example detection and robust classification. In International Conference on Learning Representations, 2020.
>
> [15] Feinman, R., Curtin, R. R., Shintre, S., and Gardner, A. B. Detecting adversarial samples from artifacts. arXiv preprint arXiv:1703.00410, 2017.
>
> [16] Miller, D., Wang, Y., and Kesidis, G. When not to classify: Anomaly detection of attacks (ada) on dnn classifiers at test time. Neural computation, 31(8):1624–1670, 2019.
>
> [17] Ma, S. and Liu, Y. Nic: Detecting adversarial samples with neural network invariant checking. In Proceedings of the 26th Network and Distributed System Security Symposium (NDSS 2019), 2019.
>
> [18] Roth, K., Kilcher, Y., and Hofmann, T. The odds are odd: A statistical test for detecting adversarial examples. In International Conference on Machine Learning, 2019.
>
> [19] Lee, K., Lee, K., Lee, H., and Shin, J. A simple unified framework for detecting out-of-distribution samples and adversarial attacks. In Advances in Neural Information Processing Systems, 2018.
>
> [20] Li, Y., Bradshaw, J., and Sharma, Y. Are generative classifiers more robust to adversarial attacks? In International Conference on Machine Learning, pp. 3804–3814. PMLR, 2019.
>
> [21] Xu, W., Evans, D., and Qi, Y. Feature squeezing: Detecting adversarial examples in deep neural networks. In Network and Distributed System Security Symposium, 2018.
>
> [22] Jha, S., Raj, S., Fernandes, S. L., Jha, S. K., Jha, S., Verma, G., Jalaian, B., and Swami, A. Attribution-driven causal analysis for detection of adversarial examples. arXiv preprint arXiv:1903.05821, 2019.
>
> [23] Hendrycks, D. and Gimpel, K. Early methods for detecting adversarial images. In International Conference on Learning Representations, 2017.
>
> [24] Yu, T., Hu, S., Guo, C., Chao, W.-L., and Weinberger, K. Q. A new defense against adversarial images: Turning a weakness into a strength. In Advances in Neural Information Processing Systems, 2019.
>
> [25] Liu, Jiayang, et al. "Detection based defense against adversarial examples from the steganalysis point of view." Proceedings of the IEEE/CVF Conference on Computer Vision and Pattern Recognition. 2019.

---

> ### Author Response · Authors · 2021-11-19
> **response to reviewer N6RH on Q3**
>
> Q3: I would also encourage the authors to include a detailed set of ablation studies, such as how many samples are needed to ensure that the surrogate models are sufficient. Does TAIG require less in comparison to others? Also, over how many samples are the Integrated Gradients computed, does that impact your results?
>
> Following your advice, we added an ablation study about the number of sampling points in TAIG-S  and TAIG-R. $\varepsilon=8/255$ and 1000 images are used in the ablation study in section A.6 in the appendix.  As suggested in [26], the number of sampling points for the study of TAIG-S starts from 20 with an increment of 10 in each of the tests.  We found that with the increase of the number of sampling points, the attack success rate of TAIG-S only slightly fluctuates. So, we stopped the sampling points at 70 and the results are listed below:
>
> | Sampling points | ResNet | Inception | PNASNet | SENet | DenseNet | MobileNet | Average |
> |:-----:|--------|-----------|---------|-------|----------|-----------|---------|
> | 20    | 100    | 25        | 15.5    | 23.7  | 50.4     | 52.8      | 33.48   |
> | 30    | 100    | 25.7      | 15.4    | 23.5  | 50.7     | 52.7      | 33.60   |
> | 40    | 100    | 25.7      | 14.6    | 25.6  | 50.8     | 52.8      | 33.90   |
> | 50    | 100    | 25.3      | 15.4    | 24.4  | 49.6     | 51.8      | 33.30   |
> | 60    | 100    | 25.8      | 14.7    | 22.8  | 50.9     | 53        | 33.44   |
> | 70    | 100    | 24.7      | 15.6    | 24    | 49.2     | 52.9      | 33.28   |
>
> For TAIG-R, we used the same setting. The number of turning points ($E$) also starts from 20 with an increment of 10 in each of the tests. And each of the line segments in the path is estimated by one sampling point. We found that increasing the number of turning points slightly improves the attack success rate. But the improvement becomes slow with the increase of the $E$. Thus, after 100 sampling points, the increment of 50 is used. The results are listed below:
>
> | $E$ | ResNet | Inception | PNASNet | SENet | DenseNet | MobileNet | Average |
> |:-----:|:------:|:---------:|:-------:|:-----:|:--------:|:---------:|:-------:|
> |   20  | 100    | 44.5      | 34.7    | 42.1  | 71.2     | 72.6      |  53.02  |
> |   30  | 100    | 45.5      | 36.2    | 44.6  | 73.1     | 73.7      |  54.62  |
> |   40  | 100    | 46.6      | 36.1    | 43.9  | 74.2     | 75        |  55.16  |
> |   50  | 100    | 47.5      | 36.4    | 45.7  | 75.7     | 75.3      |  56.12  |
> |   60  | 100    | 48.3      | 36.8    | 44.8  | 75.3     | 75.9      |  56.22  |
> |   70  | 100    | 47.8      | 37.3    | 44.5  | 75.6     | 75        |  56.04  |
> |   80  | 100    | 47.5      | 37.4    | 46.1  | 76.1     | 75.5      |  56.52  |
> |   90  | 100    | 48.3      | 37.5    | 45.7  | 76.7     | 76.1      |  56.86  |
> |  100  | 100    | 47.4      | 37.5    | 45.9  | 77.2     | 76.3      |  56.86  |
> |  150  | 100.00 |   47.70   |  37.90  | 46.00 |   76.30  |   76.40   |  56.86  |
> |  200  | 100.00 |   47.40   |  37.90  | 46.00 |   77.40  |   75.70   |  56.88  |
>
> To estimate the influence of the number of sampling points in each line segment, we fixed the turning points $E$ to 20 and changed the sampling points from 1 to 5. With the increase of the number of sampling points, the attack success rate gets slightly improvement. The results are listed below:
>
> | Sampling points| ResNet | Inception | PNASNet | SENet | DenseNet | MobileNet | Average |
> |------------|--------|-----------|---------|-------|----------|-----------|---------|
> | 1          | 100    | 44.5      | 34.7    | 42.1  | 71.2     | 72.6      | 53.02   |
> | 2          | 100    | 45.7      | 34.1    | 41.9  | 73.1     | 72.6      | 53.48   |
> | 3          | 100    | 45.6      | 35.6    | 43.9  | 74       | 74.1      | 54.64   |
> | 4          | 100    | 45.4      | 34.8    | 43.3  | 74.2     | 74        | 54.34   |
> | 5          | 100    | 46.9      | 35.2    | 44.8  | 75.5     | 74.4      | 55.36   |
>
> These experiments show that TAIG-S is not sensitive to the number of sampling points when it is larger than 20 and TAIG-R performs a bit better when the number of sampling points and the number of turning points increase.
>
> [26] Mukund, Sundararajan, Ankur Taly, and Qiqi Yan. "Axiomatic attribution for deep networks." International Conference on Machine Learning, 2017.

---

### Official Review · Reviewer_cPsc · 2021-11-02

**Correctness:** 4
**Technical Novelty And Significance:** 3
**Empirical Novelty And Significance:** 4
**Recommendation:** 8
**Confidence:** 4

**Main Review:**

Strengths:
- The application of the Integrated Gradients technique to produce adversarial examples is novel
- The evaluation of the method is comprehensive, and demonstrates state of the art performance in almost all cases

Weaknesses:
- The technical novelty is not enormous, in the sense that it consists purely of a straightforward application of an existing technique, and the method section mainly consists of a justification of why the method makes sense in this domain

**Summary Of The Paper:**

This paper applies the Integrated Gradients technique to the problem of creating adversarial examples in the black box setting. This produces results that are better than the state of the art methods for most of the networks tested, and it can be used in combination with those existing methods to produce even better results.

**Summary Of The Review:**

The paper is overall quite good, with thorough experiments demonstrating good performance.

---

> ### Public Comment · ~Zihan_Liu1 · 2021-11-13
> **wrong proof makes this paper not comprehensive**
>
> In terms of the proof of Equation 10 in appendix A.1,
>
> the authors claim that the second term is zero because of the ReLU functions in the network. However, I take a simple example that y = RELU(x1*x2). The double derivative $\frac{\partial^2y}{\partial x_1 \partial x_2} = 1$, which is not 0 when x1*x2 > 0.
>
> I believe this proof is a important one to support the methodology of this paper, which  makes the method less reliable.

---

> > ### Author Response · Authors · 2021-11-14
> > **Answer to Zihan Liu’s comment.**
> >
> > First of all, we would like to thank Zihan Liu’s comment. We use the term ReLU network to refer to the CNNs using ReLU as an activation function, which are commonly used in transferable attack experiments, instead of all networks using the ReLU activation function. If we considered all networks using the ReLU activation function, some could easily construct a function e.g., $f(x)+x^{T} x$, where f is a function with a ReLU activation function and x is a vector, with a non-zero second-order derivative. In ReLU CNN, $ReLU(w^T x)$, where $w$ is a network parameter can be considered as a basic unit and its second-order derivative is zero. Thus, the second-order derivative of the entire network is zero. In addition to our work, Wang et al. [1] also used the term ReLU network and pointed out its second-order derivative is zero. Two of their statements are given below
> >
> > “However, it is impossible for ReLU networks to run the inner-maximization due to the second-order derivatives being zeros, which is also the reason why we need Hessian approximation in this paper.”[1]
> >
> > “Direct computation of the input Hessian can be expensive and in the case of ReLU networks, not possible to optimize as the second-order derivative is zero.” [1]
> >
> > We in fact experimentally verified the statement before the submission. To avoid confusion, we will clarify the term in our revised version. Once again, thank you very much for your question.
> >
> > [1] Zifan Wang, Haofan Wang, Shakul Ramkumar, Matt Fredrikson, Piotr Mardziel and Anupam Datta, “Smoothed Geometry for Robust Attribution”, NIPS 2020.

---

> > > ### Public Comment · ~Zihan_Liu1 · 2021-11-20
> > > **Kindly Reply to Arthors**
> > >
> > > Thank you for your clarification, while I still have questions.
> > >
> > > a. As you suggested, the paper claims the traditional way is utilizing other activations like Softplus, but it is still has non-zero second derivatives.
> > > "To perform an attribution attack for ReLU networks, a common technique is to replace ReLU activations with an approximation whose second derivatives are non-zero" [1]
> > > Furthermore, the paper you refer to eliminates the second derivites due to the time comsumption, but that may cause performance loss and how could you solve this problem?
> > > Why don't you try these traditional methods and what is the difference between them?
> > >
> > > b. Your approach relies on your suggested work on adversarial attribution attack, and I think there are some issues that need theoretical explanation.
> > > First, the method is not IG, and IG's original paper does not have second derivative approximations. Why is this method implementation justified on IG?
> > > Secondly, can we verify that there is no domain gap between adversarial attribution attacks and adversarial attacks?
> > >
> > > [1] Zifan Wang, Haofan Wang, Shakul Ramkumar, Matt Fredrikson, Piotr Mardziel and Anupam Datta, “Smoothed Geometry for Robust Attribution”, NIPS 2020.

---

> > > > ### Author Response · Authors · 2021-11-24
> > > > **Response to Zihan Liu**
> > > >
> > > > Q1. As you suggested, the paper claims the traditional way is utilizing other activations like Softplus, but it still has non-zero second derivatives. "To perform an attribution attack for ReLU networks, a common technique is to replace ReLU activations with an approximation whose second derivatives are non-zero" [1] Furthermore, the paper you refer to eliminates the second derivates due to the time consumption, but that may cause performance loss and how could you solve this problem? Why don't you try these traditional methods and what is the difference between them?
> > > >
> > > > Wang et al.’s work [1] and our work have different objectives. They aim to protect attributions, but we aim to perform the transferable attack. Our proposed method does not aim to solve their performance drop due to the approximation of the Hessian matrix.
> > > > Replacing ReLU by Softplus is to compute the Hessian matrix. For ImageNet, the computational cost of the Hessian matrix is unacceptable. Using the approximated Hessian matrix [1], the Jacobian matrix of the logits w.r.t to the input is requested. For ImageNet, the dimension of the output vector is 1000. In other words, 1000 gradient calculations are needed to compute the Jacobian. This cost is still very high. In fact, in the regularization experiments (Section 5.2 [1]), Wang et al. do not examine their regularizer on ImageNet. Our attack equations are only based on 30 gradient calculations. Its computational cost is much more acceptable. However, it is no point to compare our method and Wang et al.’ work [1] because we have different objectives.
> > > >
> > > > Q. Your approach relies on your suggested work on adversarial attribution attacks, and I think there are some issues that need theoretical explanation. First, the method is not IG, and IG's original paper does not have second derivative approximations. Why is this method implementation justified on IG?
> > > >
> > > > The question is not clear to us. We try our best to answer it. We assume that you use the phrase “the method is not IG” to refer to Wang et al.’s method and the phrase “this method implementation” to refer to our method. As mentioned before, Wang et al. aims to protect the network attributions but we aim to perform the transferable attack. We use IG because IG fulfills the completeness axiom, which is useful for our analysis, is an effective attribution method for identifying key features, and performs smoothing. The last two properties are important for transferable attacks.
> > > >
> > > > Q3. Secondly, can we verify that there is no domain gap between adversarial attribution attacks and adversarial attacks?
> > > >
> > > > The formulations and objectives of adversarial attribution attacks and adversarial attacks are different. We never see anyone use the term domain gap between adversarial attribution attacks and adversarial attacks.
> > > >
> > > > [1] Zifan Wang, Haofan Wang, Shakul Ramkumar, Matt Fredrikson, Piotr Mardziel and Anupam Datta, “Smoothed Geometry for Robust Attribution”, NIPS 2020.

---

> ### Author Response · Authors · 2021-11-19
> **response to reviewer cPsc**
>
> Thanks for your positive comments.
>
> Q: The technical novelty is not enormous, in the sense that it consists purely of a straightforward application of an existing technique, and the method section mainly consists of a justification of why the method makes sense in this domain.
>
> In this paper, we proposed two different attacks (TAIG-S and TAIG-R). The TAIG-S is based on the original integrated gradients (IG) proposed by [1], which uses a straight-line path to compute the IG. To further improve the performance, we proposed a random piecewise linear path to compute the integrated gradient (RIG), which is different from the original IG in [1], and the TAIG-R attack is computed based on RIG. The adversarial examples generated by TAIG-R are proved to be more transferable than TAIG-S and the other SOTA methods. Some parts of the method section are used to pinpoint that the proposed method integrates three attack approaches into one term, which is different from other existing works.
>
>
> [1] Mukund Sundararajan, Ankur Taly, and Qiqi Yan. "Axiomatic attribution for deep networks." International Conference on Machine Learning, 2017.

---

### Official Review · Reviewer_bFo6 · 2021-11-09

**Correctness:** 2
**Technical Novelty And Significance:** 2
**Empirical Novelty And Significance:** Not applicable
**Recommendation:** 5
**Confidence:** 4

**Main Review:**

The proposed Transferable Attack based on Integrated Gradients (TAIG) generated the adversarial examples on the integrated gradients, which has shown good attack performance against some deep neural networks. The proposed attack method can be analyzed from three aspects: optimization based, attention based, and smoothing decision surface based. However, I have some concerns about this paper:
1.	It seems the paper only combines the traditional attack (FGSM) on the integrated gradients. Will the other attacks achieve better performance on the integrated gradients? More novel contributions should be stated;
2.	Why the paper selects a black image as the reference?
3.	There are still some extant works that generated the adversarial examples based on the sensitive feature in the image. It is better to describe the difference and compare the performance;
4.	I am not convinced by the explanation that sign(IG(f,x)) approximates sign(f) such that 68% elements are the same according to Fig. 2. It is better to propose any other theoretical explanation;
5.	In the experiment, the authors select thirty sampling points to estimate TAIG-S and TAIG-R. Please explain why it works?
6.	There are some typos and grammar errors in the paper, it is better to revise them.


**Summary Of The Paper:**

This paper studies adversarial attacks to deep neural network. To achieve good attack performance, this paper proposes the attack to the integrated gradients and the method (TAIG) is shown to be highly transferable to other black-box models.

**Summary Of The Review:**

The paper proposes the integrated gradients and the method (TAIG)  to attack black-box models, which has shown good performance. But there are still some place to improve.

---

> ### Author Response · Authors · 2021-11-19
> **Response to reviewer bFo6  on Q1, Q2 and Q6**
>
> We would like to thank reviewer bFO9 for the valuable and thoughtful comments. Below we address the concerns/questions mentioned in the review:
>
> Q1: The paper only combines the traditional attack on the IG? Will the other attacks achieve better performance on the IG?
>
> 1.1 In this paper, we proposed two different attacks (TAIG-S and TAIG-R). The TAIG-S is based on the original integrated gradients (IG) proposed by [1], which uses a straight-line path to compute the IG. To further improve the performance, we proposed a random piecewise linear path to compute the integrated gradients (RIG), which is different from the original IG in [1], and the TAIG-R attack is computed based on RIG. The adversarial examples generated by TAIG-R are proved to be more transferable than TAIG-S and the other SOTA methods.
>
> 1.2 We also showed that TAIG-S and TAIG-R can work together with other methods to further improve their performances. In section 4.4 of the original manuscript, we combined TAIG-S and TAIG-R with ILA, LinBP, and DI. The results are listed in Table 3. The table indicates that when combined with other methods, TAIG-S and TAIG-R can effectively enhance their attack performances. We also applied TAIG-S and TAIG-R on ILA, LinBP, and DI for target attacks. The results were given in Appendix Table 9. The table shows that TAIG-S and TAIG-R can combine with other methods to improve their performance in target setting.
>
> Q2: Why does the paper select a black image as the reference?
>
> We followed Sundararajan et al.’s suggestion [1]. They mentioned, in verbatim: “the baseline to convey a complete absence of signal, so that the features that are apparent from the attributions are properties only of the input, and not of the baseline.” In an image classification network, a black image signifies the absence of objects, so we used it as the reference.
>
> Q6: There are some typos and grammar errors in the paper, it is better to revise them.
>
> Thanks for your comments. We have carefully proofread and checked the paper. Typos and errors have been corrected.
>
>
> [1] Mukund Sundararajan, Ankur Taly, and Qiqi Yan. "Axiomatic attribution for deep networks." International Conference on Machine Learning, 2017.

---

> ### Author Response · Authors · 2021-11-19
> **Response to Reviewer bFo6 on Q3**
>
> Q3: It is better to describe the difference and compare the performance with some extant works that generated the adversarial examples based on the sensitive feature in the image.
>
> We classified the works that generate adversarial examples based on the sensitive feature as attention modification approach in Section 2. These works include AOA(2020) [4], ATA(2020) [2], TAP(2018) [5], ILA(2019) [6], and AA(2019) [3]. The proposed method uses IG and RIG to compute the attributions and exploits the attack equations, i.e., Eq. 3 and Eq. 7 to modify them. As mentioned in the paper, the proposed method integrates the three attack approaches into one single term. In other words, TAIG can be considered as a part of the attention modification approach, but it also exploits the other two approaches for achieving higher attack performance.
>
> As mentioned in the original manuscript, TAP, and ILA were not included in the evaluation because the authors of LinBP [7] showed that LinBP outperforms them significantly.  Below is the attack success rate of TAP, ILA, and LinBP under $\varepsilon=0.03$:
>
> | Methods | ResNet | Inception | PNASNet | SENet | DenseNet | MobileNet | Average |
> |:-------:|:------:|:---------:|:-------:|:-----:|:--------:|:---------:|:-------:|
> |   TAP   | 100.00 |   22.86   |  19.58  | 29.88 |   44.34  |   52.22   |  33.78  |
> |   ILA   | 100.00 |   26.20   |  28.46  | 38.52 |   53.98  |   57.74   |  40.98  |
> |  LinBP  | 100.00 |   30.80   |  29.96  | 42.74 |   63.50  |   64.18   |  46.24  |
>
>
> For AA and ATA, their authors do not share their source codes and their methods are sensitive to hyperparameters. AA is sensitive to L which indicates the layer used to compute the AA loss and ATA is sensitive to $\lambda$ which is a parameter to balance the attention loss and the global training loss. Furthermore, the authors of ATA did not clearly specify the number of layers in their ResNet V2 model. It is hard for us to make a direct comparison with them. However, Table 1 in their paper [2] indicates that ATA has similar or slightly better performance than TAP, whose performance is lower than LinBP [7]. Below is the accuracy (%) of different models under TAP and ATA attacks using ResNetV2 and InceptionV3 as the surrogate model with $\varepsilon=16/255$ given in their paper. Note that the accuracy (%) of different models in the table below is not attack success rate.
>
> | Surrogate model | Method | ResNet V2 | Inception V3 | Inception V4 | Inception-ResNet V2 |
> |-----------------|--------|-----------|--------------|--------------|---------------------|
> | ResNet V2       | TAP    | 9.5       | 51.2         | 60.1         | 55.5                |
> |                 | ATA    | 8.7       | 52.9         | 58.3         | 55.1                |
> | Inception V3    | TAP    | 48.2      | 0.1          | 24.5         | 26.3                |
> |                 | ATA    | 47.2      | 0.1          | 22.1         | 25.7                |
>
> For AA, we can only compare the results listed in their paper with the results of TAIG. The authors of AA sampled 1000 images from ImageNet and used DenseNet121 as a white-box model and ResNet50 as a black-box model. Since we do not know which 1000 images they used, we compare the number listed in their paper with our result obtained from 5000 images. The results are listed below. It should be highlighted that it is not an ideal comparison, because: (1) the authors did not share their codes, and (2) they did not specify which 1000 images they have used.
>
> | Methods      | $\varepsilon$ | ResNet50 |
> | ----------- | ----------- | ---------  |
> | AA [3]      |     0.07   |   80.58 |
> | TAIG-R   |  16/255 (0.063)  | 99.32  |
>
> [2] Wu, Weibin, et al. "Boosting the transferability of adversarial samples via attention." Proceedings of the IEEE/CVF Conference on Computer Vision and Pattern Recognition. 2020.
>
> [3] Inkawhich, Nathan, et al. "Feature space perturbations yield more transferable adversarial examples." Proceedings of the IEEE/CVF Conference on Computer Vision and Pattern Recognition. 2019.
>
> [4] Sizhe Chen, Zhengbao He, Chengjin Sun, Jie Yang, and Xiaolin Huang. Universal adversarial
> attack on attention and the resulting dataset damagenet. IEEE Transactions on Pattern Analysis
> and Machine Intelligence, 2020.
>
> [5] Wen Zhou, Xin Hou, Yongjun Chen, Mengyun Tang, Xiangqi Huang, Xiang Gan, and Yong Yang.
> Transferable adversarial perturbations. In Proceedings of the European Conference on Computer
> Vision, 2018.
>
> [6] Qian Huang, Isay Katsman, Horace He, Zeqi Gu, Serge Belongie, and Ser-Nam Lim. Enhancing
> adversarial example transferability with an intermediate level attack. In Proceedings of the IEEE
> International Conference on Computer Vision, 2019.
>
> [7] Yiwen Guo, Qizhang Li, and Hao Chen. Backpropagating linearly improves transferability of adversarial examples. In Advances in Neural Information Processing Systems, 2020.

---

> ### Author Response · Authors · 2021-11-19
> **Response to reviewer bFo6 on Q5**
>
> Q5: Please explain why select thirty sampling points to estimate TAIG-S and TAIG-R and why it works?
>
> Sundararajan et al. [1] pointed out that the sampling points between 20 to 300 are enough to approximate the integral and we followed their advice. To balance the computation cost and attack success rate, we used 30 sampling points in the experiments.
>
> To further investigate the influence of the number of sampling points, we did an ablation study under $\varepsilon=8/255$ using 1000 images and added it to the appendix in Section A.6. As suggested in [1], the number of sampling points for TAIG-S is tested from 20 with an increment of 10 in each of the tests.  We found that with the increase of the number of sampling points increasing, the attack success rate of TAIG-S only slightly fluctuates. Therefore, we tested it up to 70 and the results are listed below:
>
> | Sampling points | ResNet | Inception | PNASNet | SENet | DenseNet | MobileNet | Average |
> |:-----:|--------|-----------|---------|-------|----------|-----------|---------|
> | 20    | 100    | 25        | 15.5    | 23.7  | 50.4     | 52.8      | 33.48   |
> | 30    | 100    | 25.7      | 15.4    | 23.5  | 50.7     | 52.7      | 33.60   |
> | 40    | 100    | 25.7      | 14.6    | 25.6  | 50.8     | 52.8      | 33.90   |
> | 50    | 100    | 25.3      | 15.4    | 24.4  | 49.6     | 51.8      | 33.30   |
> | 60    | 100    | 25.8      | 14.7    | 22.8  | 50.9     | 53        | 33.44   |
> | 70    | 100    | 24.7      | 15.6    | 24    | 49.2     | 52.9      | 33.28   |
>
> For TAIG-R, we used the same setting. The number of turning points ($E$) is set to 20 at the start with an increment of 10 in each of the tests. After $E=100$, the step size increases to 50. (Because the improvement is very slow). And each of the line segments in the path is estimated by one sampling point. We found that increasing the number of turning points slightly improves the attack success rate. But the improvement is very minor when $E$ is larger than 50. The results are listed below:
>
> | $E$ | ResNet | Inception | PNASNet | SENet | DenseNet | MobileNet | Average |
> |:-----:|:------:|:---------:|:-------:|:-----:|:--------:|:---------:|:-------:|
> |   20  | 100    | 44.5      | 34.7    | 42.1  | 71.2     | 72.6      |  53.02  |
> |   30  | 100    | 45.5      | 36.2    | 44.6  | 73.1     | 73.7      |  54.62  |
> |   40  | 100    | 46.6      | 36.1    | 43.9  | 74.2     | 75        |  55.16  |
> |   50  | 100    | 47.5      | 36.4    | 45.7  | 75.7     | 75.3      |  56.12  |
> |   60  | 100    | 48.3      | 36.8    | 44.8  | 75.3     | 75.9      |  56.22  |
> |   70  | 100    | 47.8      | 37.3    | 44.5  | 75.6     | 75        |  56.04  |
> |   80  | 100    | 47.5      | 37.4    | 46.1  | 76.1     | 75.5      |  56.52  |
> |   90  | 100    | 48.3      | 37.5    | 45.7  | 76.7     | 76.1      |  56.86  |
> |  100  | 100    | 47.4      | 37.5    | 45.9  | 77.2     | 76.3      |  56.86  |
> |  150  | 100 |   47.70   |  37.90  | 46.00 |   76.30  |   76.40   |  56.86  |
> |  200  | 100 |   47.40   |  37.90  | 46.00 |   77.40  |   75.70   |  56.88  |
>
> To estimate the influence of the number of sampling points in each line segment, we fixed the turning points $E$ to 20 and changed the sampling points from 1 to 5. With the increase in the number of sampling points, the attack success rate improves slightly. The results are listed below:
>
> | Sampling points| ResNet | Inception | PNASNet | SENet | DenseNet | MobileNet | Average |
> |------------|--------|-----------|---------|-------|----------|-----------|---------|
> | 1          | 100    | 44.5      | 34.7    | 42.1  | 71.2     | 72.6      | 53.02   |
> | 2          | 100    | 45.7      | 34.1    | 41.9  | 73.1     | 72.6      | 53.48   |
> | 3          | 100    | 45.6      | 35.6    | 43.9  | 74       | 74.1      | 54.64   |
> | 4          | 100    | 45.4      | 34.8    | 43.3  | 74.2     | 74        | 54.34   |
> | 5          | 100    | 46.9      | 35.2    | 44.8  | 75.5     | 74.4      | 55.36   |
>
> These experiments show that TAIG-S is not sensitive to the number of sampling points when it is larger than 20 and TAIG-R performs a bit better when increasing the number of sampling points or the number of turning points.
>
> [1] Mukund Sundararajan, Ankur Taly, and Qiqi Yan. "Axiomatic attribution for deep networks." International Conference on Machine Learning, 2017.

---

> ### Author Response · Authors · 2021-11-19
> **Response to reviewer bFo6  on Q4**
>
> Q4: I am not convinced by the explanation that sign(IG(f,x)) approximates sign(f) such that 68% elements are the same according to Fig. 2. It is better to propose any other theoretical explanation;
>
> We believe that the reviewer focuses on the number, 68%, and the word, approximation. First of all, we would like to highlight that the mathematical discussion is not to prove that $\mathrm{sign}({\nabla f})$ and $\mathrm{sign}(\boldsymbol{\mathrm{IG}}(\boldsymbol{x}))$  are highly similar. It is to indicate that they have some similarity and where their difference comes from. As mentioned in the original paper, “the quality of this approximation depends on Eq. 12”, which is
> \begin{equation}
> 	\frac{\partial\mathrm{IG}_i(\boldsymbol{x})}{\partial x_i}\approx\frac{\mathrm{IG}_i(\boldsymbol{x})-\mathrm{IG}_i(\boldsymbol{x}-h\Delta\boldsymbol{x})}{h}$
> \end{equation}
> Let us draw an analogy between the first-order Taylor approximation and our mathematical discussion. It would be easier to explain it. The first-order Taylor approximation is given below
>
> $g(x) \approx g(a)+g^{'}(a)(x-a).$
>
> It is well-known that the quality of this approximation depends on the difference between $x$ and $a$. When $a$ is far away from $x$, the quality of the approximation would drop. As with the first-order Taylor approximation, the quality of our approximation (Eq. 12) depends on $h$, which indicates the difference between $x$ and $x-h \Delta x$. In the derivation, we do not define $h$ explicitly because $\mathrm{IG}_i (x-h \Delta x)$ is set to zero. More clearly, the issue about the quality of approximation in Eq. 12 is the same as the first-order Taylor approximation. When two points are closer, the quality of the approximation will be better. However, to perform a transferable attack, $\mathrm{IG}_i (x-h \Delta x)$ is set to zero, which influences the quality of the approximation. It should be highlighted that a better approximation is not our goal. When $\mathrm{sign}(\boldsymbol{\mathrm{IG}}(\boldsymbol{x}))$  is very close to $\mathrm{sign}({\nabla f})$, it only implies that it is stronger for white-box attack. However, it is well-known that $\mathrm{sign}({\nabla f})$ is not ideal for the black-box attack, which is the target of this study. More clearly, we do not expect that $\mathrm{sign}(\boldsymbol{\mathrm{IG}}(\boldsymbol{x}))$  is very close to $\mathrm{sign}({\nabla f})$. To avoid confusion, in the revised version, remarks have been added and the term weakly approximate is used whenever is suitable.

---

### Author Response · Authors · 2021-11-23
**Update summary**

Dear reviewers, thanks for your valuable comments. We have made some major updates based on your comments. Here is a quick summary:

Revision:

+ Section 2.3: we added the references of Admix and VT.
+	Section 3.2: we changed the name of the attacks to “Transferable Attack using” to “Transferable Attack based on”.
+	Section 3.3: we used the term weakly approximates wherever is suitable and added a footnote to clarify that the goal of the proposed methods is not to achieve better approximation.
+	Section 4.1: we modified the description of TAIG-R to make the setting of parameter $E$ and sampling points clearer.
+	Appendix A.1: we clarified the term ReLU network and added a footnote for it.

New experimental results:

+	Appendix A.3.1: we added the experiments of LinBP and SI with the same number of gradient calculations as TAIG-S and TAIG-R. (Table 8)
+	Appendix A.3.2: we added the experiments to compare TAIG-R with LinBP on VGG19. (Table 9 and Table 10).
+	Appendix A.3.3: we added the experiments to compare TAIG-R with NI-VT, MI-VT, and MI-Admix. (Table 11 and Table 12)
+ Appendix A.5: we added the experiments of two adversarial image detectors. (Table 15)
+	Appendix A.6: we added the ablation study about the sampling points for TAIG-S and the number of turning points and sampling points for TAIG-R. (Table 16, 17, and 18)
+    Appendix A.7: we added the perceptual evaluation of different methods. (Table 19)

We hope the updated manuscript could address your concerns.

---

### Decision · Program_Chairs · 2022-01-20

**Decision:**

Accept (Poster)

**Comment:**

This paper proposes integrating three existing approaches to give a simple algorithm called TAIG for generating transferable adversarial examples under blackbox attacks.

In the original reviews, some strengths and weaknesses of the papers were highlighted although some of them have not reached general agreement after the discussion period.

Regarding the merits, it is generally felt that the experimental results are good and the idea of updating along the integrated gradients is new (despite a simple idea) and has some theoretical justification.

Nevertheless, even after the discussion period, some concerns still remain, including the technical novelty of the proposed method and the high computational requirements of the proposed method, among others.

We appreciate the authors for responding to the reviews by clarifying some points and providing further experimental results. The paper would be more ready for publication if all the comments and suggestions are taken into consideration to improve the paper more thoroughly.